# Untangling the Mistral and seasonal atmospheric forcing driving deep convection in the Gulf of Lion: 2012-2013

Douglas Keller Jr.[1], Yonatan Givon[2], Romain Pennel[1], Shira Raveh-Rubin[2], and Philippe Drobinski[1]

[1]LMD/IPSL, École Polytechnique, Institut Polytechnique de Paris, ENS, PSL Research University, Sorbonne Université, CNRS, Palaiseau, France
[2]Department of Earth and Planetary Sciences, Weizmann Institute of Science, Rehovot, Israel

**Correspondence:** Douglas Keller Jr. (dg.kllr.jr@gmail.com)

**Abstract.** Deep convection in the Gulf of Lion is believed to be primarily driven by the Mistral winds. However, our findings show that the seasonal atmospheric change provides roughly 2/3 of the buoyancy loss required for deep convection to occur, for the 2012 to 2013 year, with the Mistral supplying the final 1/3. Two NEMOMED12 ocean simulations of the Mediterranean Sea were run for the Aug. 1st, 2012 to July 31st, 2013 year, forced with two sets of atmospheric forcing data from a RegIPSL coupled run within the Med-CORDEX framework. One set of atmospheric forcing data was left unmodified, while the other was filtered to remove the signal of the Mistral. The Control simulation featured deep convection, while the Seasonal did not. A simple model was derived, relating the anomaly scale forcing (the difference between the Control and Seasonal runs) and the seasonal scale forcing to the ocean response through the Stratification Index. This simple model revealed that the Mistral's effect on buoyancy loss depends more on its strength rather than its frequency or duration. The simple model also revealed that the seasonal cycle of the Stratification Index is equal to the net surface heat flux over the course of the year, with the stratification maximum and minimum occurring roughly at the fall and spring equinoxes.

## 1 Introduction

Deep convection, also known as open-ocean convection, is an important ocean circulation process that typically occurs in the high latitude regions (Marshall and Schott, 1999). Localized events are triggered by the reduction of the stable density gradient through sea surface layer buoyancy loss. One such area of deep convection is the Gulf of Lion (GOL) in the Mediterranean Sea. The deep convection events that occur in this region aid the general thermohaline circulation of the Mediterranean Sea by forming the Western Mediterranean Dense Water (WDMW) (Robinson et al., 2001). After its formation, this dense water spreads out along the northwestern basin, among the deeper layers of the Med. Sea (MEDOC, 1970), with some transported along the northern boundary current towards the Balearic Sea (Send and Testor, 2017), and some transported to the south within eddies (Beuvier et al., 2012; Testor and Gascard, 2003) into the southern Algerian Basin and towards the Strait of Gibraltar (Béranger et al., 2009), completing the cyclonic circulation pattern of the sea. The water column mixing that occurs during

a deep convection event also brings oxygenated water down from the oxygen-rich sea surface layer and injects sea-bottom nutrients upwards towards the surface (Coppola et al., 2017; Severin et al., 2017), resulting in increased phytoplankton blooms in the following season (Severin et al., 2017).

Significant deep convection events occur every few years in the GOL (Somot et al., 2016; Houpert et al., 2016; Marshall and Schott, 1999; Mertens and Schott, 1998), driven by the Mistral and Tramontane winds. These sister northerly flows bring cool, continental air through the Rhône Valley (Mistral) and the Aude Valley (Tramontane) leading to large heat transfer events with the warmer ocean surface (Drobinski et al., 2017; Flamant, 2003). These cooling and evaporation events destabilize the water column in the GOL, and are widely accepted to be the primary source of buoyancy loss leading to deep convection (Lebeaupin-Brossier et al., 2017; Houpert et al., 2016; L'Hévéder et al., 2012; Lebeaupin-Brossier et al., 2012; Herrmann et al., 2010; Lebeaupin-Brossier and Drobinski, 2009; Noh et al., 2003; Marshall and Schott, 1999; Mertens and Schott, 1998; Madec et al., 1996; Schott et al., 1996; Madec et al., 1991b,a; Gascard, 1978).

Here, we investigated the Mistral's role in deep convection in the GOL (as the Mistral and Tramontane winds are sister winds, we will refer to them jointly as "Mistral" winds). It's role was determined by running two NEMO ocean simulations of the Med. Sea from Aug. 1st, 2012 to July 31st, 2013, forming a case study of the encapsulated winter. One simulation was forced by unmodified atmospheric forcing data, while the other was forced by a filtered atmospheric dataset with the signal of the Mistral removed from the forcing. Thus, the ocean response due to the Mistral events could be separated and examined, revealing the effects of seasonal atmospheric change alone. A multitude of observational data was collected during this year in the framework of the HYdrological cycle in the Mediterranean EXperiment (HyMeX) (Estournel et al., 2016b; Drobinski et al., 2014), which provided a solid base of observations to validate the ocean model results.

In particular, our findings quantify:

- the separated and combined effect of the Mistral and seasonal atmospheric cycle on deep convection,

- the dominant attribute of the Mistral causing buoyancy loss,

- the source of the buoyancy loss due to the seasonal atmospheric cycle.

Two additional years were also studied with the same methodology as the 2012-2013 winter: the 1993-1994 and 2004-2005 winter. The 1993-1994 winter does not have a deep convection event, and allows us to compare a deep convecting year versus a non deep convecting year. The 2004-2005 winter is a well studied deep convecting winter and offers some additional literature to draw analysis upon, as well as an additional deep convecting year to compare and contrast with, using the same methodology as the 2012-2013 winter.

There are three distinct sections of the deep convection cycle: the preconditioning phase in the fall, the main, large over-turning phase in the winter and early spring (when deep convection occurs), and the restratification/spreading phase during the proceeding summer (MEDOC, 1970; Group, 1998). The focus of study is on the preconditioning and overturning phase where the Mistral is stronger and more frequent (Givon et al., 2021) and therefore plays a larger role in the deep convection cycle.

The model used and the methodology is described in the methods section (Sec. 2). Model results and validation are presented in the results and discussion section for the 2012-2013 winter (Sec. 3). Patterns observed in the model results lead to the

development of a simple model that describes the Mistral's and seasonal cycle's effect. This simple model is presented in the following process analysis section (Sec. 4). The two additional years, 1993-1994 and 2004-2005 are presented in the comparison with additional years section (Sec. 5), proceeding the two former sections focusing on the 2012-2013 winter. Our concluding remarks are presented in the conclusions section (Sec. 6).

## 2 Methodology

In our study we used the NEMO ocean model to run two ocean simulations forced by unmodified and modified atmospheric forcing data from a coupled WRF/ORCHIDEE simulation. Information on the Mistral events, used later when developing the simple model in Sec. 4, was extracted from the unmodified atmospheric forcing data and from ERA Interim Reanalysis data. The main metric used in this article to examine the model results and relate them to deep convection is the Stratification Index, $SI$. Each of these components are described below in their own subsection.

### 2.1 NEMO

The Nucleus for European Modelling of the Ocean (NEMO) ocean model (https://www.nemo-ocean.eu/) was used in bulk formula configuration to simulate the GOL region with two distinct simulations, both performed from Aug. 1st, 2012 to July 31st, 2013. In bulk formula configuration, sea surface fluxes are computed from parameterized formulas using atmospheric and oceanic measurable variables as inputs, such as temperature and wind velocity. The following parameterized formulas are used to calculate the latent heat flux, $Q_E$, the sensible heat flux, $Q_H$, the longwave radiation heat flux, $Q_{LW}$, and the surface shear stress, $\tau$:

$$
\begin{aligned}
Q_E &= \rho_0 \Lambda C_E (q_z - q_0)|\Delta \boldsymbol{u}| \\
Q_H &= \rho_0 c_p C_H (\theta_z - SST)|\Delta \boldsymbol{u}| \\
Q_{LW} &= Q_{LW,a} - \epsilon \sigma SST_K^4 \\
\tau &= \rho_0 C_D \Delta \boldsymbol{u} |\Delta \boldsymbol{u}|
\end{aligned}
\tag{1}
$$

where $z$ is the height above the sea surface the atmospheric variables are provided at, with the naught values ($_0$) representing the values at the sea surface. $\boldsymbol{u}$ is the horizontal wind vector, with $\Delta \boldsymbol{u} = \boldsymbol{u}_z - \boldsymbol{u}_0$ as the difference between the wind velocity and sea surface current (assuming a no-slip condition at the ocean surface). $q$ and $\theta$ are the specific humidity and potential temperature of air, respectively. $\Lambda$ and $c_p$ are the latent heat of evaporation and the specific heat of water, respectively. $\rho_0$ is the air density at the sea surface. $SST$ is the sea surface temperature, with $SST_K$ as the sea surface absolute temperature. $\epsilon$ is the sea surface emissivity, $\sigma$ is the Stefan-Boltzmann constant, and $Q_{LW,a}$ is the atmospheric longwave radiation. $C_E$, $C_H$, and $C_D$ are the coefficients of latent heat, sensible heat, and drag, respectively, and are defined in Large and Yeager (2004, 2008).

The net downward heat flux, $Q_{net}$, is described by the summation of the terms in the following equation (Large and Yeager, 2004; Estournel et al., 2016b):

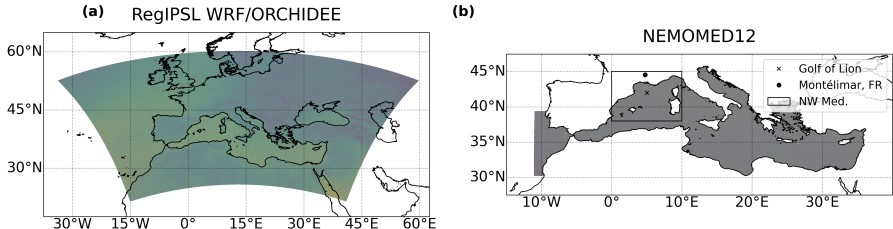

**Figure 1.** The domains of both the WRF domain from the RegIPSL coupled WRF/ORCHIDEE simulation within the Med-CORDEX framework, (a), and the NEMOMED12 configuration domain, (b). The region of interest, the NW Med., is outlined by the box. This region is later used in Fig. 7. The location used to study the temporal development of deep convection in the GOL is at $42°$ N $5°$ E, and the other location, used in conjunction with the aforementioned point to determine Mistral events, is Montélimar, FR, at $44.56°$ N $4.75°$ E.

$$Q_{net} = Q_{SW} + Q_{LW} + Q_H + Q_E \qquad (2)$$

where $Q_{SW}$ is the downward shortwave radiation. Snowfall and precipitation are included in the simulation calculations but excluded here for brevity in the following sections, as the buoyancy loss due to the water flux at the surface is essentially negligible (Somot et al., 2016).

The NEMO model was run in the NEMOMED12 configuration using NEMO v3.6. The domain is shown in Fig. 1 (b); it covers the Mediterranean Sea and a buffer zone representing the exchanges with the Atlantic Ocean. This configuration features

a horizontal resolution of $1/12°$ (roughly 7km) and 75 vertical levels (with a variable vertical resolution from 1m at the surface to 135m at the bottom). The 3-D temperature and salinity fields are restored towards the ORAS4 global ocean reanalysis (Balmaseda et al., 2013) in the buffer zone. The conservation of volume in the buffer zone is achieved through strong damping of the sea surface height (SSH) towards the ORAS4 reanalysis. The Black Sea, runoff of 33 major rivers, and coastal runoff is represented by climatological data input from Ludwig et al. (2009). A deeper explanation of the configuration and boundary

conditions is given by the works Waldman et al. (2018); Hamon et al. (2016); Beuvier et al. (2012); Lebeaupin-Brossier et al. (2011); Arsouze et al. (2012). The initial conditions were provided by an ocean objective analysis by Estournel et al. (2016b).

### 2.2 Atmospheric forcing

The atmospheric forcing used in the simulations were the outputs of RegIPSL, the regional climate model of IPSL (Guion et al., 2021), which used the coupling of the Weather Research and Forecasting Model (WRF) (Skamarock et al., 2008) and

the ORCHIDEE Land Surface Model (Krinner et al., 2005). The run was a hind-cast simulation (ERA interim downscaling), performed at 20 km resolution, spanning the period of 1979-2016, within the Med-CORDEX framework (Ruti et al., 2016). The $u$ and $v$ components, specific humidity, potential temperature, shortwave and longwave downward radiation, precipitation, and snowfall were all used to force the ocean simulations.

For the "Control" simulation, the forcing were used as is. For the "Seasonal" simulation, the $u$ and $v$ wind components, specific humidity, and potential temperature were filtered, see Fig. 2, over the entire domain shown in Fig. 1 (a). These variables were chosen as they are the primary variables that affect the surface flux calculations in the bulk formulae (Eq. set (1)). The variables relating to radiation and precipitation fluxes were left unchanged. The filtering removes the short term, anomaly scale forcing from the forcing dataset (the phenomena with under a month timescale), effectively removing the Mistral's influence on the ocean response. This creates two separate forcing datasets, one with the anomaly scale forcing included, one with just the seasonal scale forcing (hence the designation of Control and Seasonal).

The filtering process was performed by a moving window average:

$$\chi_i = \frac{1}{i+N+1} \sum_{j=0}^{i+N} x_j \tag{3}$$

where $\chi_i$ is the averaged (filtered) value at index $i$ of a time series of variable $x$ with length $n$, where $i = 0 \to n$. The window size is equal to $2N+1$, which, in this case, is equal to 31 days. The ends have a reduced window size for averaging, and thus show edge effects. The edge effects did not affect the forcing used for the NEMO simulations, as they were before and after the ocean simulation dates, as two, full year atmospheric forcing data were used for the simulations.

The moving window average was applied to each time point per day over a 31 day window (i.e. for 3 hourly data, the time series is split into 8 separate series, one for each timestamp per day - 00:00, 03:00, 06:00, etc. - then averaged with a moving window before being recombined). This was done to retain the average intra-day variability yet smooth the intra-monthly patterns, as the diurnal cycle has been shown to retard destratification by temporarily reforming a stratified layer at the sea surface during slight daytime warming. This diurnal restratification has to be overcome first before additional destratification of the water column can continue during the next day (Lebeaupin-Brossier et al., 2012, 2011) and is shorter than the typical Mistral event length of about 5.69 days (Table 1).

An important note must be made about the filtering process. The Mistral primarily acts in the higher frequency range but at a lower frequency than the diurnal cycle, as mentioned above. However, it also features signal strength in the lower frequencies on the seasonal scale. This is due to the fact the Mistral becomes stronger, longer, and more frequent during the preconditioning phase than during the rest of the annual cycle (Givon et al., 2021). The moving window averaging we have applied to filter the Mistral out of the atmospheric forcing primarily removes the higher frequency portion of the Mistral's presence. But it also removes part of the lower frequency portion as well, that other filters, such as the Butterworth filter, struggle with, without removing more of the seasonal signal than intended that isn't influenced by the Mistral. This reveals a very interesting point about the structure of the Mistral and the use of "seasonal" and "anomaly" timescales. Since the Mistral primarily acts in the higher frequencies, the "anomaly" timescale will refer to both the higher and lower frequency portions of the Mistral that were filtered out, but will be treated mainly as referring to the higher frequency portion in discussion. The "seasonal" timescale will then refer to the remaining lower frequency signal and average diurnal cycle post filtering.

The result of the filtering is shown in Fig. 2. Temperature and specific humidity were filtered as is, while the wind speed, was component ($u$ and $v$) filtered, preserving the general wind direction (Fig. 2 wind direction polar plot). Due to the slow

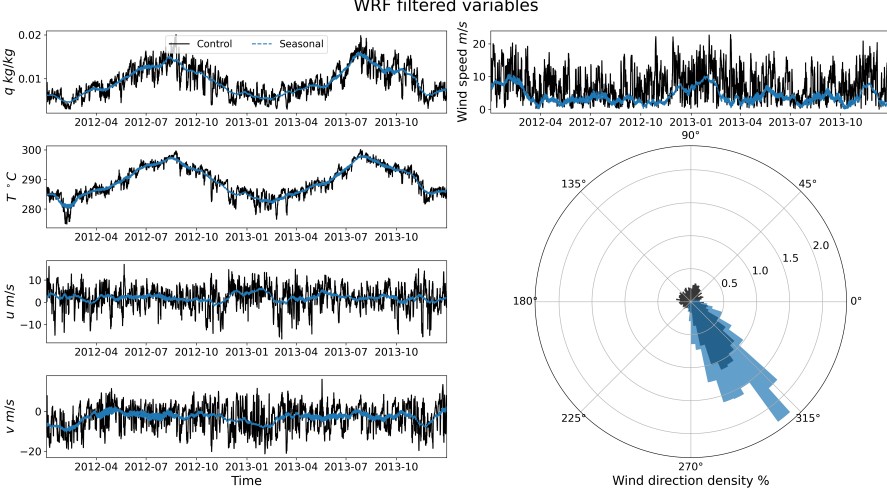

**Figure 2.** An illustration of the filtering (averaging) process described by Eq. (3). Here the variables $q$, $T$, $u$, and $v$ are shown for both the unfiltered (Control; black) and filtered (Seasonal; blue) datasets at the nearest grid point to 42° N 5° E. Note how the peaks of the time series are removed and the general wind direction is conserved.

movement of intermediate and dense water, which is on the order of about a year scale for newly formed WMDW to move into the southern Algerian Basin (Beuvier et al., 2012) and on the order of decades for total circulation (Millot and Taupier-Letage, 2005), we assume the processes outside the NW Med. subdomain in Fig. 1 (b), that are affected by the filtering have a negligible

impact on the GOL processes on the preconditioning phase timescale.

## 2.3 Mistral events

Mistral events will be used for developing the simple model in the process analysis section (Sec. 4), for their role in driving buoyancy loss at the ocean surface. Events were determined from the WRF/ORCHIDEE dataset, in combination with the ERA Interim Reanalysis dataset (Dee et al., 2011). Two main criteria were used to define a Mistral event:

1. Northerly flow with a stream-wise flow direction $\pm 45°$ about the south cardinal direction, above $2\ m/s$ at two locations simultaneously: at Montélimar, France (45.5569° N, 4.7495° E) and in the GOL (42.6662° N, 4.4372° E).

2. The presence of a Genoa Low, defined as a closed sea level pressure contour around a minimum in the field, using 0.5 hPa intervals, anywhere in the box defined by the latitudes 38 and 44° N and longitudes 4 and 14° E (a slightly different domain than that of Givon et al. (2021)).

The events during the preconditioning period, Aug. 30th, 2012 to Feb. 21st, 2013, were then manually checked and edited to remove single day gaps to better represent the data according to a visual inspection of the atmospheric forcing data. For $k$ Mistral events, each event's duration, $\Delta t_k$, and period from the beginning of the event to the next event, $\Delta \tau_k$, was determined

**Table 1.** The start date of, duration of, $\Delta t_k$, and period between each event, $\Delta \tau_k$, for each Mistral event, $k$, for the entire NEMO simulation period of Aug. 1st, 2012 to July 31st, 2013. Superscripts $d$ and $a$ denote events used as ideal cases for calculating $\alpha_d$ and $\alpha_a$, respectively, in Sec. 4.3 and App. A2.1 and A2.2.

| Start Date | $\Delta t_k$ $days$ | $\Delta \tau_k$ $days$ | Start Date | $\Delta t_k$ $days$ | $\Delta \tau_k$ $days$ |
|---|---|---|---|---|---|
| 2012-08-03 | 1 | 3 | 2012-12-26 | 5 | 7 |
| 2012-08-06 | 1 | 2 | 2013-01-02 | 17 | 21 |
| 2012-08-08 | 1 | 5 | 2013-01-23 | 6 | 10 |
| 2012-08-13 | 1 | 12 | 2013-02-02$^d$ | 15 | 18 |
| 2012-08-25 | 2 | 5 | 2013-02-20 | 7 | 10 |
| 2012-08-30$^{d,a}$ | 8 | 13 | 2013-03-02 | 1 | 11 |
| 2012-09-12$^{d,a}$ | 4 | 7 | 2013-03-13 | 3 | 7 |
| 2012-09-19$^{d,a}$ | 2 | 9 | 2013-03-20 | 1 | 6 |
| 2012-09-28$^{d,a}$ | 5 | 14 | 2013-03-26 | 1 | 5 |
| 2012-10-12$^{d,a}$ | 4 | 15 | 2013-03-31 | 1 | 6 |
| 2012-10-27$^d$ | 5 | 15 | 2013-04-06 | 2 | 13 |
| 2012-11-11$^{d,a}$ | 3 | 8 | 2013-04-19 | 4 | 8 |
| 2012-11-19 | 2 | 8 | 2013-04-27 | 1 | 25 |
| 2012-11-27$^d$ | 6 | 11 | 2013-05-22 | 2 | 10 |
| 2012-12-08$^d$ | 4 | 9 | 2013-06-01 | 2 | 23 |
| 2012-12-17$^d$ | 3 | 4 | 2013-06-24 | 1 | 4 |
| 2012-12-21 | 2 | 5 | 2013-06-28 | 1 | 41 |

The average values for Mistral events from 2012-08-30 to 2013-02-16 are: $\overline{\Delta t} = 5.69\ days$ and $\overline{\Delta \tau} = 10.88\ days$. The standard deviations for the same time frame are: $\sigma_{\Delta t} = 4.22\ days$ and $\sigma_{\Delta \tau} = 4.59\ days$.

and is provided in Table 1 for the entire ocean simulation period (for further analysis into the selection of this criteria, see Givon et al. (2021)).

## 2.4 Stratification Index

A useful metric to quantify the vertical stratification of a column of water is the Stratification Index, $SI$ (Léger et al. (2016); Somot et al. (2016); Somot (2005); sometimes called the "convection resistance"). It's derived from the non-penetrative growth of the Mixed Layer Depth (MLD; i.e. without entrainment; Turner (1973)), which has been shown to be an accurate approximation for open ocean convection (Marshall and Schott, 1999):

$$\frac{\partial z}{\partial t} = \frac{B(t)}{N^2(z)z} \tag{4}$$

where $N^2$ is the Brunt-Väisälä frequency, $z$ is the vertical coordinate along the water column, $\frac{\partial z}{\partial t}$ is the growth of the mixed layer depth, and $B$ is the potential buoyancy loss the water column can endure before removing stratification (in units of $m^2/s^3$). Separating by variables and integrating results in the equation for $SI$:

$$SI = \int_0^D N^2 z \, dz \tag{5}$$

where $D$ is the depth of water column. If $N^2$ is assumed to be constant throughout the water column, the integral simplifies to:

$$SI = \frac{D^2}{2} N^2 \tag{6}$$

$SI$ provides a 0 dimensional index to track stratification and can be easily related to the buoyancy loss experienced by the water column due to the atmosphere. Because of this, in this article $SI$ will be used as the diagnostic to track the atmosphere's impact on the stratification of the GOL waters.

## 3 Winter of 2013

The results are presented in two parts for the 2012-2013 winter (additionally referred as the 2013 winter): model validation against observational data and the model results of the deep convection cycle presented from the center of convection, roughly at 42°N 5°E. This section and the next (Sec. 4), primarily focus on the winter of 2013.

### 3.1 Model validation

To validate the model results, data from the HyMeX (https://www.hymex.org/) database was compared to the NEMO control simulation. Sea surface temperature (SST) data from Météo-France's Azur and Lion buoy were compared with the Control simulation SST of the nearest grid point in NEMOMED12. Figure 3 shows the comparison. The Azur buoy data was missing SST measurements from Jan. 19th, 2013 to July 10th, 2013, but where the data is available, NEMO corresponds well to the observations. The same is true for the Lion buoy data, which had measurements for the entire time covered by the simulations. This comes as no surprise, as the NEMOMED12 simulations' SST is restored to the observational dataset of Estournel et al. (2016b). However, this also means that the calculated surface sensible heat fluxes should be fairly accurate, as both the sensible heat flux and latent heat flux calculations depend on the SST (Eq. (1)).

Additionally, the Control simulation density and potential temperature profiles were compared to Conductivity-Temperature-Depth (CTD) measurements also procured from the HyMeX database. The CTD measurements were collected during the HyMeX Special Observation Period 2 (Taupier-Letage Isabelle, 2013; Estournel et al., 2016a; Drobinski et al., 2014) mission. The CTD profiles collected at approximately the same time and location were averaged together to adjust for small variances

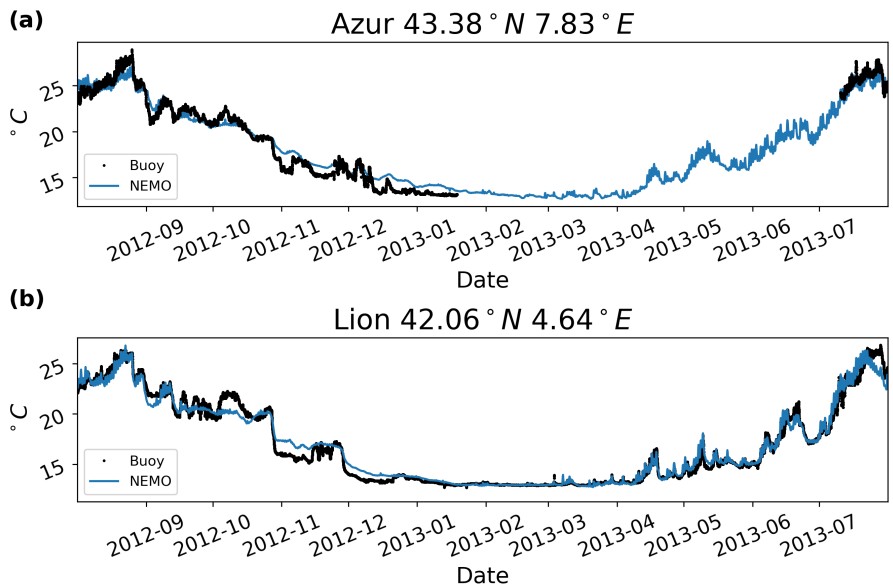

**Figure 3.** SST comparison between the NEMO Control run and the Azur , (a), and Lion, (b), buoy SST datasets. Where the data is available, the model results match the buoy data fairly well.

and gaps in the data. The averaged profiles and their standard deviations are visualized in Fig. 5 and Fig. 6. The locations of the CTD profiles are shown in Fig. 4.

Like with the SST comparisons, the profiles from the nearest grid point in the Control simulation domain were used for the CTD comparisons. The Root Mean Square Error (RMSE) and bias (calculated as the difference between the model values and the observation values) for each of the averaged CTD profiles and corresponding Control simulation profiles was calculated and is presented in Table 2. Overall, the Control simulation and CTD profiles are decently well correlated but not perfect, with low RMSE and bias for both density and potential temperature. The density profiles have an average RMSE less than the average RMSE for the potential temperature profiles: 0.025 $kg/m^3$ and 0.094 $°C$, respectively.

Argo float profiles from the HyMeX database were also compared to the Control simulation, again with profiles from the nearest grid point being used. 3118 potential temperature profiles within the box bounded by the 40 to 44° N latitudes and the 2 to 8° longitudes, to represent the GOL area, were considered (see Fig. 4). The average RMSE between the Argo profiles and Control simulation profiles was 0.43 $°C$, with an average bias of 0.23 $°C$. These values are larger than the values of the comparison with the CTD profiles. However, considering the shear volume of profiles and, during stratified conditions the temperature can range a few degrees from the surface to the lower layers, these results aren't unexpected.

Temperature differences on the order of $10^{-2}$ $°C$ are potentially all that is required to sustain an ocean convective cycle (Marshall and Schott, 1999) and density differences for the same order of magnitude, $10^{-2}$ $kg/m^3$, are used to separate newly formed dense water during deep convection (Houpert et al., 2016; Somot et al., 2016; Beuvier et al., 2012). This means

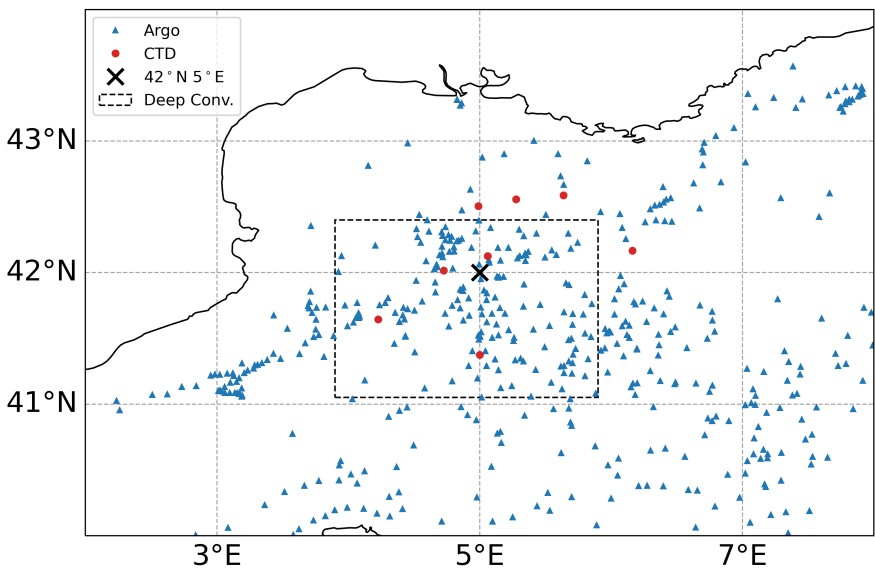

**Figure 4.** Locations of the CTD and Argo profiles. The red circles represent the CTD locations and the blue triangles represent the Argo float profile locations. The deep convection area is marked by the box with a dashed perimeter and 42° N 5° E is marked by an "X".

our model results should be studied with a critical eye, as they may not be fully representative of the true ocean response, given the bias and RMSE values from comparing the simulation to CTD and Argo profiles. Additionally, meanders around 40km in wavelength form due to baroclinic instability along edge of the convection patch (Gascard, 1978). This could mean the deviations from observations are due to out-of-phase meanders around the convective patch region in the model relative to actuality. Regardless, we believe the simulations are accurate enough to provide interesting results for the transient and regional scale response of the GOL, which covers the main interest of our study.

### 3.2 Stratification Index and Mixed Layer Depth

Figure 7 shows the $SI$ calculated over the GOL for both simulations: row (a) for the Control and row (b) for the Seasonal. An important distinction between the two results is deep convection is present in the Control simulation but not the Seasonal. This is more clearly seen in Fig. 8 (c) (closest NEMOMED12 grid point to 42° N, 5° E), as the Control simulation MLD reaches the sea floor on Feb. 13, 2013, while the Seasonal MLD remains close to the sea surface. This confirms that atmospheric forcing with timescales less than a month, e.g. the Mistral winds, provide a significant amount of buoyancy loss, as without them deep convection fails to occur. There is, however, still significant loss of stratification at the location of the GOL gyre in the Seasonal simulation, which is visible in row (b) of Fig. 7 on the date of Feb. 13th, 2013. This spot of destratification is present, but less so, in the preceding and proceeding plots of the same row.

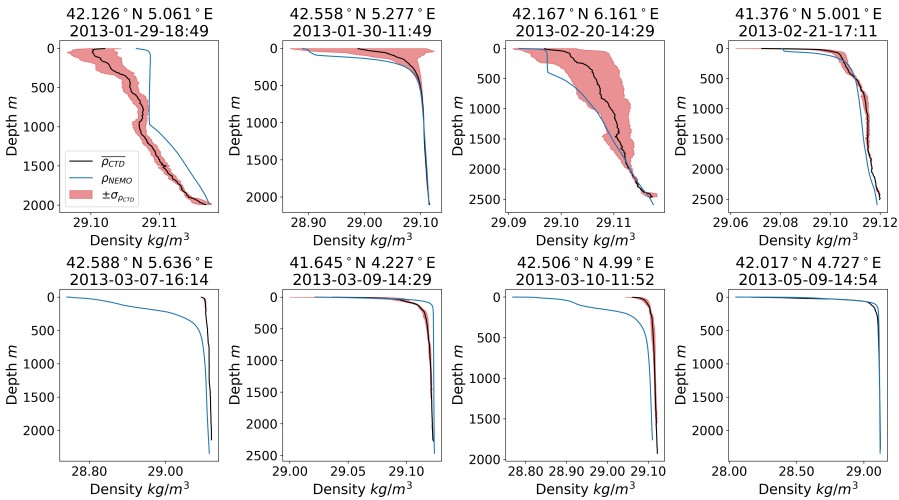

**Figure 5.** Comparison of CTD and NEMO Control simulation density profiles. The CTD profiles were averaged by combining multiple vertical profiles collected at the date and location into one profile. The standard deviation of this averaging, $\sigma_{\rho_{CTD}}$, is marked in red and is present for all plots, yet may be difficult to see for March 7th and May 9th.

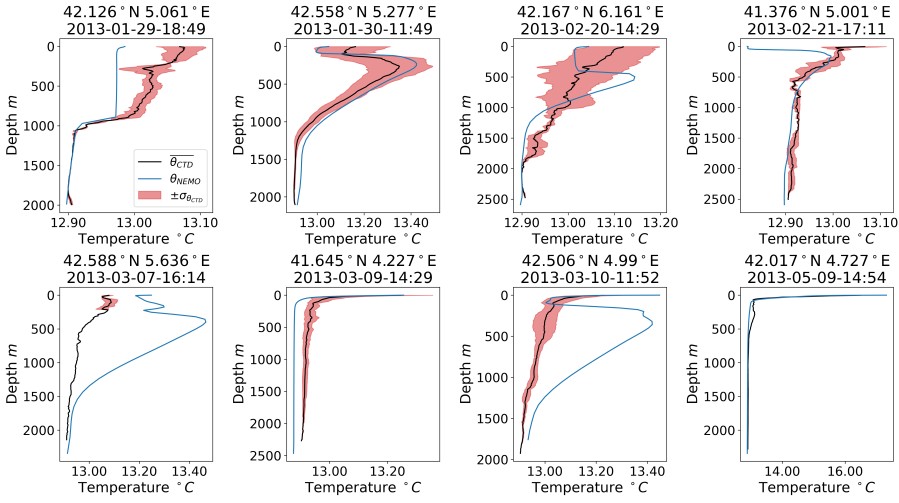

**Figure 6.** Same as Fig. 5 but for potential temperature.

To investigate the time series ocean response in more detail, a spatially averaged time series of the $SI$ for both simulations was analyzed at the grid point nearest to $42°$ N, $5°$ E. These coordinates were selected as it is the point with the most destratification in Fig. 7, and is the typical center of deep convection in the GOL (Marshall and Schott, 1999; MEDOC, 1970). The spatial averaging involved horizontally averaging the immediately adjacent grid points, such that 9 grid points in total were averaged, centered around $42°$ N, $5°$ E. The Stratification Index from the Control simulation is given as the sum of $\delta SI + SI_S$,

**Table 2.** RMSE and bias between the averaged observed CTD density and potential temperature profiles and the nearest NEMO Control grid point profiles, for the respective variables.

| Date | Lat. $deg$ | Lon. $deg$ | RMSE$_\rho$ $kg/m^3$ | RMSE$_\theta$ $^\circ C$ | Bias$_\rho$ $kg/m^3$ | Bias$_\theta$ $^\circ C$ |
|---|---|---|---|---|---|---|
| 2013-01-29-18:49 | 42.126 | 5.061 | 0.004 | 0.041 | 0.0032 | -0.0265 |
| 2013-01-30-11:49 | 42.558 | 5.277 | 0.030 | 0.055 | -0.0093 | 0.0348 |
| 2013-02-20-14:29 | 42.167 | 6.161 | 0.004 | 0.050 | -0.0026 | -0.0059 |
| 2013-02-21-17:11 | 41.376 | 5.001 | 0.003 | 0.033 | -0.0019 | -0.0120 |
| 2013-03-07-16:14 | 42.588 | 5.636 | 0.077 | 0.233 | -0.0377 | 0.1751 |
| 2013-03-09-14:29 | 41.645 | 4.227 | 0.005 | 0.043 | 0.0036 | -0.0414 |
| 2013-03-10-11:52 | 42.506 | 4.990 | 0.058 | 0.224 | -0.0320 | 0.1719 |
| 2013-05-09-14:54 | 42.017 | 4.727 | 0.018 | 0.077 | 0.0046 | -0.0367 |

The average RMSE and bias for the density profiles was 0.025 $kg/m^3$ and -0.009 $kg/m^3$, respectively. The average RMSE and bias for potential temperature was 0.094 $^\circ$C and 0.032 $^\circ$C, respectively.

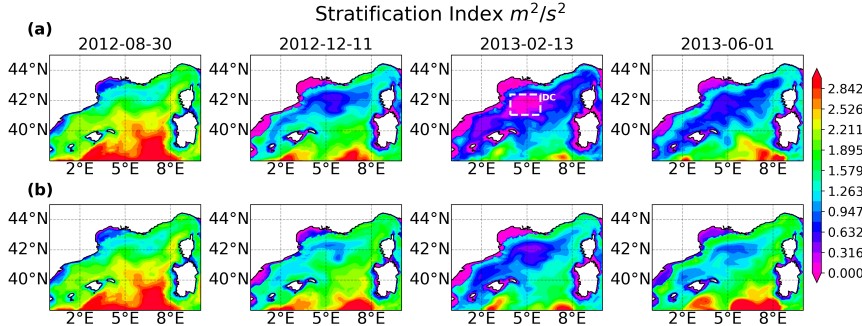

**Figure 7.** The Stratification Index across the GOL (the area marked as NW Med. in Fig. 1 (b)) at different timestamps. Row (a) displays the values of $SI$ for the Control simulation and row (b) displays the values of $SI$ for the Seasonal simulation. The box denoted by $DC$ indicates the area of deep convection in the GOL that was not seen in the Seasonal simulation.

while the Stratification Index of the Seasonal simulation is given as $SI_S$. The difference between the two, $\delta SI$, should contain the change in stratification due to shorter timescale atmospheric events, such as the Mistral, because of the filtering performed in Sec. 2.2. $\delta SI + SI_S$, $SI_S$, and $\delta SI$ are all shown in Fig. 8.

Both the Control and Seasonal runs start off with an $SI$ value of 1.57 $m^2/s^2$ (beginning of Fig. 8 (a)), then diverge at the first major Mistral event starting on August 30th, 2012. After diverging, the two runs remain diverged until the end of the simulation run time, ending with a difference of about -0.22 $m^2/s^2$, which is seen in $\delta SI$ (shown in Fig. 8 (b)). As commented earlier, the most striking difference between the Control and Seasonal run is the occurrence of deep convection in the Control run, occurring when $\delta SI + SI_S$ is equal to 0 (signified also when the MLD reaches the sea floor), and the lack of deep convection in the Seasonal run, as $SI_S$ only reaches a minimum of 0.43 $m^2/s^2$. Additionally, if only the anomaly timescale atmospheric

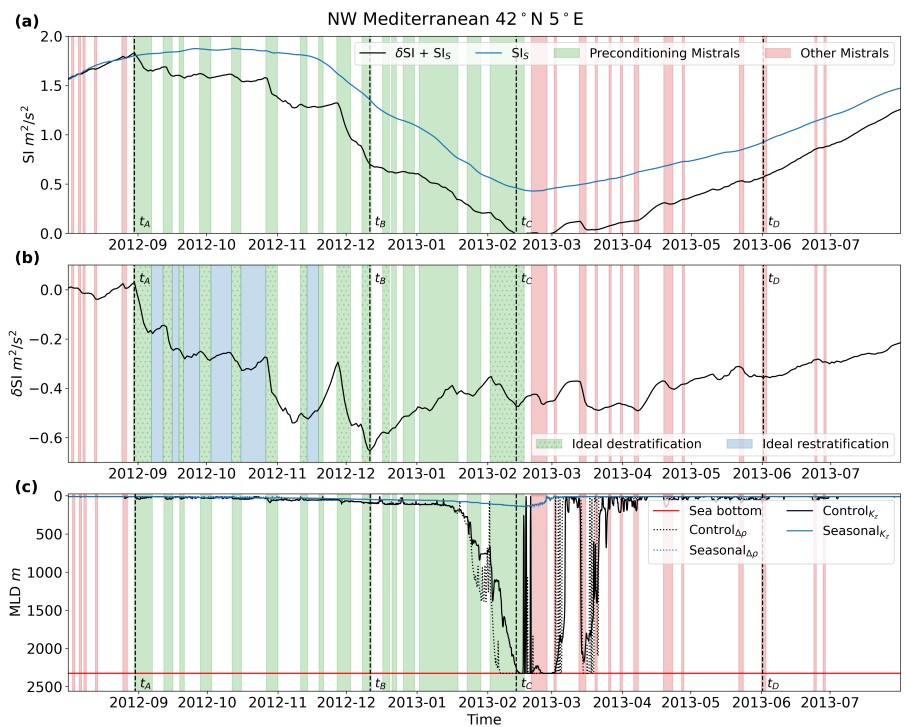

**Figure 8.** The Stratification Index of the nearest NEMO grid point to $42°$ N $5°$ E and MLD over the year of both simulations. Plot (a) shows the Stratification Index for the Control run, $SI_S + \delta SI$, and the Seasonal run, $SI_S$. Plot (b) shows the difference between the Control and Seasonal Stratification Index, $\delta SI$. Plot (c) shows the MLD for both simulations. Mistral events are shown in all three plots: colored green for events during the preconditioning and deep convection phase and red for events outside of the preconditioning phase. Mistral events with dotted hatching (the blue colored intervening time between events) are used as ideal destratification (restratification) events to compute the simple model restoration coefficients. The specific timestamps $t_A$ through $t_D$ correspond to the timestamps of the plots in Fig. 7: Aug. 30th, 2012, Dec. 11th 2012, Feb. 13th, 2013, June 1st, 2013, respectively. Two definitions of MLD are plotted in (c): one calculated by a vertical change in density less than $0.01\ kg/m^3$, denoted by $\Delta\rho$, and one calculated by a vertical diffusivity less than $5 \times 10^{-4}\ m^2/s$, denoted by $K_z$. The MLD denoted by the vertical diffusivity criteria follows the turbocline depth and is taken to represent the mixed layer depth more accurately, as it matches the deep convection timing in the Stratification Index.

forcing is considered, hence $\delta SI$ is the only stratification change from the initial $1.57\ m^2/s^2$, the roughly $-0.6\ m^2/s^2$ of
maximum destratification that the anomaly timescale provides is not enough to overcome the initial stratification. This means that both the intra-monthly and the inter-monthly variability of the buoyancy loss, reflected in $\delta SI + SI_S$, are required for deep convection to occur.

Another significant result is the timing of the deep convection. Deep convection initially occurs on Feb. 13th, 2013, which is before $SI_S$ reaches its minimum on Feb. 21th, 2013, but after $\delta SI$ reaches its minimum on Dec. 11th, 2012. After $\delta SI$
reaches its minimum, it stays around $-0.43\ m^2/s^2$ until May 2013, where it starts to increase. This means that while the

induced destratification from the anomaly scale forcing would have been able to overcome $0.6\ m^2/s^2$ of stratification to form deep convection in Dec., the seasonal stratification was only low enough in Feb. for both $\delta SI$ and $SI_S$ to have a combined destratification strong enough for the water column to mix. In other words, the seasonal atmospheric forcing destratified the already preconditioned water column into deep convection along with a simultaneous Mistral event. This means buoyancy loss due to the anomaly forcing may not necessarily be the only trigger for deep convection, at least for this year. This can be seen more clearly in the MLD, as the MLD grows over two Mistral events preceding it reaching the seafloor.

## 4  Process analysis

To pick apart how the atmospheric forcing influences the stratification in the Gulf of Lion, a simple model was developed to separate out the individual components of interest for both the seasonal and anomaly time scales.

### 4.1  Simple model derivation

To connect the Mistral to the ocean's response, we make the assumption that the response is a superposition of the seasonal response and the anomaly response. This means the effects of the Mistral can be categorized as anomalies affecting the short term anomaly timescale and studied separately from the seasonal response. In terms of the Brunt-Väisälä frequency, this linear combination is represented by $N^2 = \delta N^2 + N_S^2$, where $\delta$ denotes the anomaly terms and $S$ denotes the seasonal terms. To determine the Mistral's effect, we derive a simple model to describe the $SI$ of the water column in response to atmospheric forcing (**note:** the full derivation is found in App. A1). We start with the energy equation for incompressible fluids (White, 2011), then multiply the equation by $-g/T_0$ to express the energy equation in terms of buoyancy, assuming that the ocean's density varies negatively proportionally with temperature, $\rho = -\beta T$. We then perform partial differentiation with respect to $z$ to obtain an equation describing the Brunt-Väisälä frequency, $N^2$, in response to the atmospheric forcing, given by a forcing function, $F(t)$. Separating by timescale, we arrive at the following partial differential equations:

$$\begin{aligned} \frac{D\delta N^2}{Dt} &= -\delta F(t) \\ \frac{DN_S^2}{Dt} &= -F_S(t) \end{aligned} \tag{7}$$

$F(t)$ is preceded by a minus sign for ease in derivation, as positive quantities of $F(t)$ mean heat, hence buoyancy, is removed from the water column.

To simplify the seasonal time scale, we assume $N_S^2$ is only a function of time, $t$:

$$\frac{dN_S^2}{dt} = -F_S(t) \tag{8}$$

Assuming a homogeneous seasonal Brunt-Väisälä frequency over the depth of the water column gives us the relation for the seasonal Stratification Index, $SI_S$, and seasonal atmospheric forcing:

$$\frac{\mathrm{d}SI_S}{\mathrm{d}t} = -\frac{D^2}{2}F_S(t) \tag{9}$$

To simplify the analytical solution for the anomaly timescale, we describe the advection term, $\boldsymbol{V} \cdot \nabla(\delta N^2)$, as a restoring

term, $R = \alpha(\delta N^2)$, which relates the overall Brunt-Väisälä frequency to its seasonal component, $N_S^2$, using the linear assumption made before. This means the restoring coefficient, $\alpha$, represents the advective operation. This results in the following differential equation:

$$\frac{\partial \delta N^2}{\partial t} + \alpha(\delta N^2) = -\delta F(t) \tag{10}$$

### 4.2 Seasonal solution and forcing

The solution for the seasonal timescale is relatively straight forward. As shown before, Eq. (9) relates the seasonal stratification, $SI_S$, to the seasonal atmospheric forcing, $F_S(t)$. We have the following definition of $F_S(t)$ from App. A1:

$$F_S(t) = \frac{\partial}{\partial z}\left(\frac{\mathbf{q_{a,S}}g}{\rho c_p T_0}\right) = \frac{g}{\rho c_p T_0}\frac{\partial \mathbf{q_{a,S}}}{\partial z} \tag{11}$$

where $c_p$ is the specific heat capacity of water, taken as $4184\ Jkg^{-1}K^{-1}$, $g$ is gravity, $\rho$ is the density of water, taken as $1000$ $kgm^{-3}$, and $T_0$ is the reference temperature, taken as the average seasonal sea surface temperature of $292.4\ K$. This means

$SI_S$ can be related to the seasonal volumetric atmospheric heat transfer, $\mathbf{q_{a,S}}$. Setting $\mathbf{q_{a,S}} = -Q_{net,S}/D$, where $Q_{net,S}$ is the seasonal net downward heat flux at the ocean surface from Eq. (2), we can calculate $\frac{\mathrm{d}SI_S}{\mathrm{d}t}$ from $Q_{net,S}$. If we integrate both sides of Eq. (9) by $z$, after plugging in Eq. (11) and the relationship for $Q_{net,S}$, as $SI_S$ is constant with respect to (w.r.t.) $z$, Eq. (9) becomes:

$$\frac{\partial SI_S}{\partial t} = \frac{g}{2\rho c_p T_0}Q_{net,S} \tag{12}$$

$\frac{g}{2\rho c_p T_0} \approx 10^{-9}\ m^4/Js^2$, which means the derivative of $SI_S$ w.r.t. time, $t$, multiplied by $10^9$ is on the same order of magnitude as $Q_{net}$ (with the subscript $S$ now dropped for convenience, as the rest of the subsection discusses seasonal heat fluxes), which is what we see in Fig. 9 (a) for the 2013 winter, with $\frac{\mathrm{d}SI_S}{\mathrm{d}t} \times 10^9$ following the curve of $Q_{net}$. This relationship means when $Q_{net}$ crosses zero with a negative derivative, $SI_S$ experiences a maximum and vice versa for a minimum. Additionally, the longer $Q_{net}$ remains negative, the more seasonal destratification is incurred by the ocean. The seasonal variation of $Q_{net}$ is

primarily driven by the solar radiation, $Q_{SW}$, which is evident in Fig. 9 (b). Consequently, the maximum and minimum values for $SI_S$ occur around Sept. 21st and March 21st, the fall and spring equinoxes. The asymmetry in $Q_{net}$ is mostly caused by the slightly seasonally varying latent heat flux, $Q_E$, followed by the sensible heat flux, $Q_H$, both of which also decrease the net heat flux by roughly $100\ W/m^2$ to $200\ W/m^2$, depending on the time of the year. $Q_{LW}$ remains roughly constant during

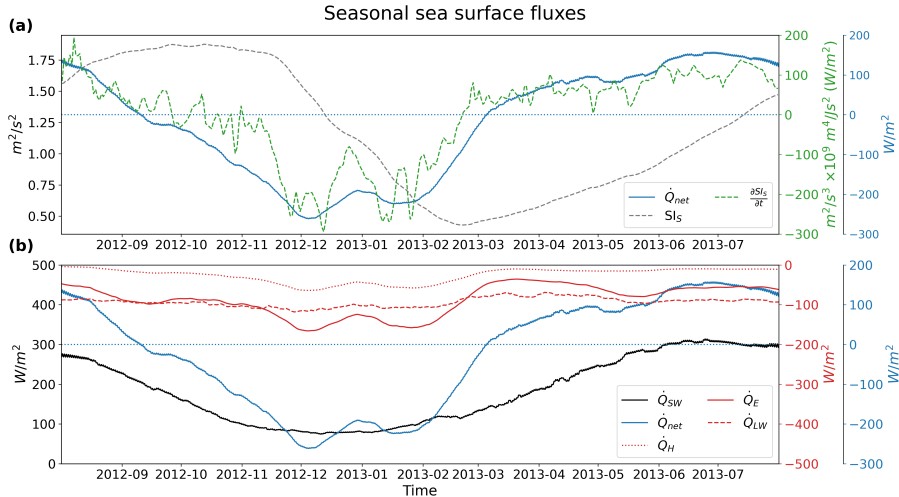

**Figure 9.** The smoothed (with Eq. (3)) seasonal surface heat fluxes over the point $42°$ N $5°$ E for the Seasonal simulation. (a) contains the seasonal stratification index, $SI_S$, and its derivative, $\frac{\partial SI_S}{\partial t}$, comparing it to the seasonal net heat flux, $Q_{net}$ (the subscript $S$ is dropped for convenience). (b) shows the net heat flux separated into its components: $Q_E$, $Q_H$, $Q_{SW}$, and $Q_{LW}$ for latent heat, sensible heat, shortwave downward, and longwave downward fluxes, respectively (neglecting contributions from precipitation and snowfall). The different line colors correspond to the similarly colored axes.

the year, decreasing $Q_{net}$ by roughly -100 $W/m^2$. These results are corroborated by the results of multiple model reanalysis

for the region as well (Song and Yu, 2017).

Equation (12) and Fig. 9 convey that the seasonal stratification is primarily driven by shortwave downward radiation. The other terms, the longwave, latent heat, and sensible heat, shift the net heat flux negative enough for the ocean to have a destratification/restratification cycle. If the net heat flux was always positive, stratification would continue until the limit of the simple model applicability. This is an important finding, as, if future years feature less latent and sensible heat exchange due to

warming or more humid winters, there will be less seasonal destratification, requiring more destratification from the anomaly timescale to cause deep convection. Consecutive years of decreasing latent and sensible heat fluxes could form a water column that is too stratified to allow deep convection to occur.

### 4.3 Anomaly solution and forcing

To solve for the anomaly timescale, described by Eq. (10), we assume $\delta F(t)$ can be represented by a pulse function shown in

Fig. 10. This pulse function assumes the primary forcing at the anomaly timescale is represented by the Mistral events. Each Mistral event, $k$, has a duration, $\Delta t_k$, and a period between the start of the current and following event, $\Delta \tau_k$. $\delta F_k$ is the strength of the forcing for each event. Inserting this into Eq. (10) allows us to solve it in a piecewise manner. Like what we did for the seasonal time scale, we assume the water column has a homogeneous Brunt-Väisälä frequency, allowing us to make use of Eq. (6). The restoring coefficient then only represents the horizontal advection, as the vertical component becomes zero with our

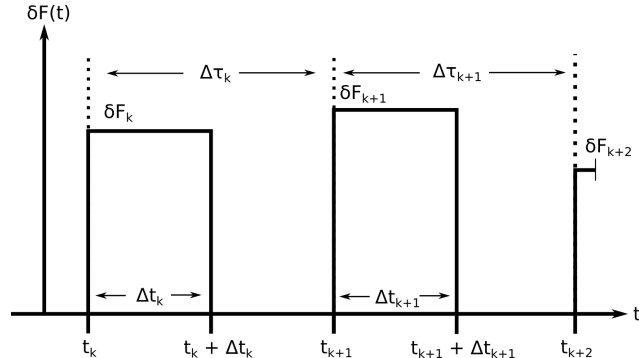

**Figure 10.** The Mistral forcing as a pulse function used to solve Eq. (10). $k$ corresponds to the event and $\delta F_k$ corresponds to the forcing strength of the Mistral event. $\Delta t_k$ corresponds to the duration of the of the Mistral event, and $\Delta \tau_k$ to the period between events, with $t_k$ denoting the start of event $k$.

assumption of a homogeneous $N^2$. The last assumption is the restoring coefficient remains constant for each section of the forcing function:

$$\delta SI_k(t) = \begin{cases} \left[\delta SI_{k-1}(t_k) + \frac{\delta F_k}{\alpha_d}\left(1 - e^{\alpha_d(t-t_k)}\right)\right] e^{-\alpha_d(t-t_k)} & [t_k, t_k + \Delta t_k) \\ \left[\delta SI_{k-1}(t_k) + \frac{\delta F_k}{\alpha_d}\left(1 - e^{\alpha_d \Delta t_k}\right)\right] e^{(\alpha_a - \alpha_d)\Delta t_k - \alpha_a(t-t_k)} & [t_k + \Delta t_k, t_k + \Delta \tau_k) \end{cases} \quad (13)$$

where $\alpha_d$ and $\alpha_a$ are the restoring coefficients during ($[t_k, t_k + \Delta t_k)$) and after ($[t_k + \Delta t_k, t_k + \Delta \tau_k)$) a Mistral event, respectively.

Further assuming $\delta F_k = \delta F$, $\Delta t_k = \Delta t$, and $\Delta \tau_k = \Delta \tau$ for all $k$, which results in a periodic pulse function with constant amplitude and period, we can simplify Eq. (13) using the sum of a finite geometric series. At the beginning of the preconditioning period, destratification hasn't yet begun, therefore the initial $\delta SI$ is zero, resulting in the following equation set:

$$\delta SI_k(t) = \begin{cases} \frac{D^2}{2}\frac{\delta F}{\alpha_d}\left[\left(1 - e^{\alpha_d \Delta t}\right)\left(\frac{1 - e^{[(\alpha_a - \alpha_d)\Delta t - \alpha_a \Delta \tau]k}}{1 - e^{[(\alpha_a - \alpha_d)\Delta t - \alpha_a \Delta \tau]}} - 1\right) + \left(1 - e^{\alpha_d(t-t_k)}\right)\right] e^{-\alpha_d(t-t_k)} & [t_k, t_k + \Delta t_k) \\ \frac{D^2}{2}\frac{\delta F}{\alpha_d}\left[\left(1 - e^{\alpha_d \Delta t}\right)\left(\frac{1 - e^{[(\alpha_a - \alpha_d)\Delta t - \alpha_a \Delta \tau]k}}{1 - e^{[(\alpha_a - \alpha_d)\Delta t - \alpha_a \Delta \tau]}} - 1\right) + \left(1 - e^{\alpha_d \Delta t_k}\right)\right] e^{(\alpha_a - \alpha_d)\Delta t_k - \alpha_a(t-t_k)} & [t_k + \Delta t_k, t_k + \Delta \tau_k) \end{cases}$$

$$(14)$$

This final equation set allows us to describe the integrated effect of consecutive Mistrals and to easily pick apart the effects of the Mistral's different attributes, including the frequency of the events.

To determine the value of the restoring coefficients, a normalized function was derived for each section of a Mistral event (derivation shown in App. A2.1 for during an event and App. A2.2 for after an event). The resulting normalized functions were fitted against the NEMO $\delta SI$ results in Fig. 8 for the denoted ideal events in Table 1 (denoted d for the dates with ideal destratification taking place during the event and a for the dates with ideal restratification taking place after the event) and

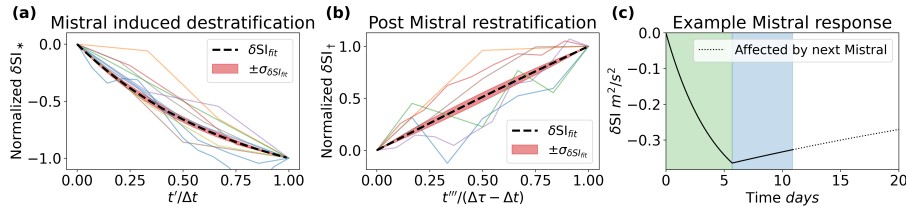

**Figure 11.** The normalized theoretical solutions (Eq. (A42) and (A48)) for during, (a), and after, (b), a destratification event fitted to the ideal Mistral events from Table 1 and $\delta SI$ values from the NEMO results in Fig. 8. A value of 0.235 $day^{-1}$ for $\alpha_d$ and a value of 0.021 $day^{-1}$ for $\alpha_a$ was found. Plot (c) shows the $\delta SI$ response using the determined restoration coefficients, given an ideal Mistral event with the average values of 5.69 $days$ for the duration and 10.88 $days$ for the period. The average strength of a Mistral, $\delta F = 4.01 \times 10^{-8} \; s^{-2} days^{-1}$, was taken from values found in Table A1 from the Appendix.

given the average event values of $\overline{\Delta t} = 5.69 \; days$ and $\overline{\Delta \tau} = 10.88 \; days$. The result of the fitting is shown in Fig. 11, with $\alpha_d$ having a fitted valued of 0.235 $day^{-1}$ and $\alpha_a$ having a fitted value of 0.021 $day^{-1}$. If we recall the meaning of $\alpha_d$ and $\alpha_a$ from the derivation of the simple model in Sec. 4.1, this means the advective term in Eq. (10) has a larger role in the destratification phase of the Mistral event than in the restratification phase, as it is an order of magnitude larger. This result suggests horizontal mixing occurs between events, as a smaller value for the restoration coefficient during the restratification phase means the existence of weaker horizontal gradients than during the preceding destratification phase.

The strength of each Mistral event, $\delta F_k$, was found in a similar way by solving for $\delta F_k$ after noting the initial value of $\delta SI_k(t_k)$ is equal to $\delta SI_{k-1}(t_k)$ (derivation found in App. A3). Then the values of $\delta SI$ from the NEMO results in Fig. 8 were plugged in to determine the values of $\delta F_k$ (see Table A1 in the appendix for the resulting values).

### 4.3.1 Mistral strength and destratification

Mistral events do not always lead to destratification. Some events in Fig. 8 fail to create further destratification and actually continue to restratify the water column. The simple model can describe this phenomena. To determine which events lead to destratification versus not, we take the derivative with respect to time of Eq. (13) for during an event. This results in the following equation:

$$\frac{\partial \delta SI_k(t)}{\partial t} = -\alpha_d \left[ \delta SI_{k-1}(t_k) + \frac{D^2}{2} \frac{\delta F_k}{\alpha_d} \right] e^{-\alpha_d(t-t_k)} \tag{15}$$

The quantity $\frac{\partial \delta SI_k(t)}{\partial t}$ must be less than zero for destratification to occur, which means if $\alpha_d$ is a positive quantity (refer to App. A2 or Fig. 11), $\delta SI_{k-1}(t_k) + \frac{D^2}{2} \frac{\delta F_k}{\alpha_d}$ must be a positive quantity. If some destratification has already occurred relative to the seasonal stratification, such that $\delta SI_{k-1}(t_k) < 0$, then $\frac{D^2}{2} \frac{\delta F_k}{\alpha_d}$ must be larger than $-\delta SI_{k-1}(t_k)$ for destratification to occur. Recalling that $\delta F_k$ is positive when heat is removed from the water column, this means that additional Mistral events must overcome the current amount of destratification to further destratify the water column. Otherwise, no destratification occurs or even restratification occurs. An example of this can be seen with the Mistral event starting on Jan. 2nd, 2013, that

lasts for 17 days in Fig. 8 (b). The event starts off with an initial destratification of -0.48 $m^2/s^2$ and ends at -0.41 $m^2/s^2$, a net restratification of 0.07 $m^2/s^2$. This is despite the fact this event has a positive $\delta F_k$ value of $3.80 \times 10^{-8} \ s^{-2} day^{-1}$ (from Table A1).

The combined overall effect of this result can be seen in Fig. 8 (b), as the consecutive Mistral events during the precondi-
350 tioning phase cause destratification to a minimum of -0.6 $m^2/s^2$ for $\delta SI$ on Dec. 11th, 2012. Proceeding events after this minimum fail to continue to destratify the water column and, instead, restratification occurs on the anomaly time scale, even before deep convection occurs. The seasonal stratification, $SI_S$, and along with the anomaly destratification, $\delta SI$, brings the total $SI$ to zero on Feb. 13, 2013, resulting in deep convection.

### 4.3.2 Dominating Mistral attribute

A pertinent question to ask is which attribute of the Mistral, the frequency, strength, or duration, is the most important when it drives destratification. Figure 13 and 12 show the results of varying $\delta F$, $\Delta t$, and $\Delta \tau$ individually (in subplots (a), (b), and (c)), respectively) in Eq. (14). The other variables are kept at the mean value when not varied. The dashed lines in both figures show the limit of potential destratification per case. What we can see is stronger Mistral events, with an increased value for $\delta F$, result in more destratification, with the reverse happening with decreased values. Decreasing the event duration, $\Delta t$, results
in less destratification, however, increasing event duration causes more destratification up to the limit where the individual events converge into one single long event and the destratification converges to the dashed line limit. After this, there is no additional destratification. Increasing or decreasing the frequency of events (decreasing or increasing the period, $\Delta \tau$), only minimally changes the accrued destratification, due to the fact that the magnitude of $\frac{\partial \delta SI}{\partial t}$ is dependent on the strength of the current Mistral event and the already achieved destratification. Decreasing the frequency (increasing the period), allows for
more restratification to occur after an event, but the proceeding event has a larger difference between current destratification and the event strength, leading to destratification that almost reaches the same level as the case with more frequent events. Increasing the frequency has a similar effect to increasing the duration; when the period is zero, the forcing becomes one large event, converging the resulting destratification to the dashed line.

To more accurately quantify the effect of each attribute, we separate $\delta SI$ into its total derivative in terms of the Mistral
attributes:

$$\mathrm{d}\delta SI = \underbrace{\frac{\partial \delta SI}{\partial \delta F}\mathrm{d}\delta F}_{\text{Strength}} + \underbrace{\frac{\partial \delta SI}{\partial \Delta t}\mathrm{d}\Delta t}_{\text{Duration}} + \underbrace{\frac{\partial \delta SI}{\partial \Delta \tau}\mathrm{d}\Delta \tau}_{\text{Period}} \tag{16}$$

Due to the lack of available total derivatives for $\delta F$, $\Delta t$, and $\Delta \tau$, we approximate them with their respective standard deviation: $\sigma_x \approx \mathrm{d}x$. Before we determine the partial derivatives for each attribute, note that in Fig. 12 and 13 subplot $f$ that changing the number events, $k$, does not change the potential destratification limit (the dashed line). This means the potential
destratification does not change with the number of events. Another notation to make is the character of the potential destratification limit: it approaches some asymptotic value as $k$ approaches infinity. We can take advantage of this by differentiating the

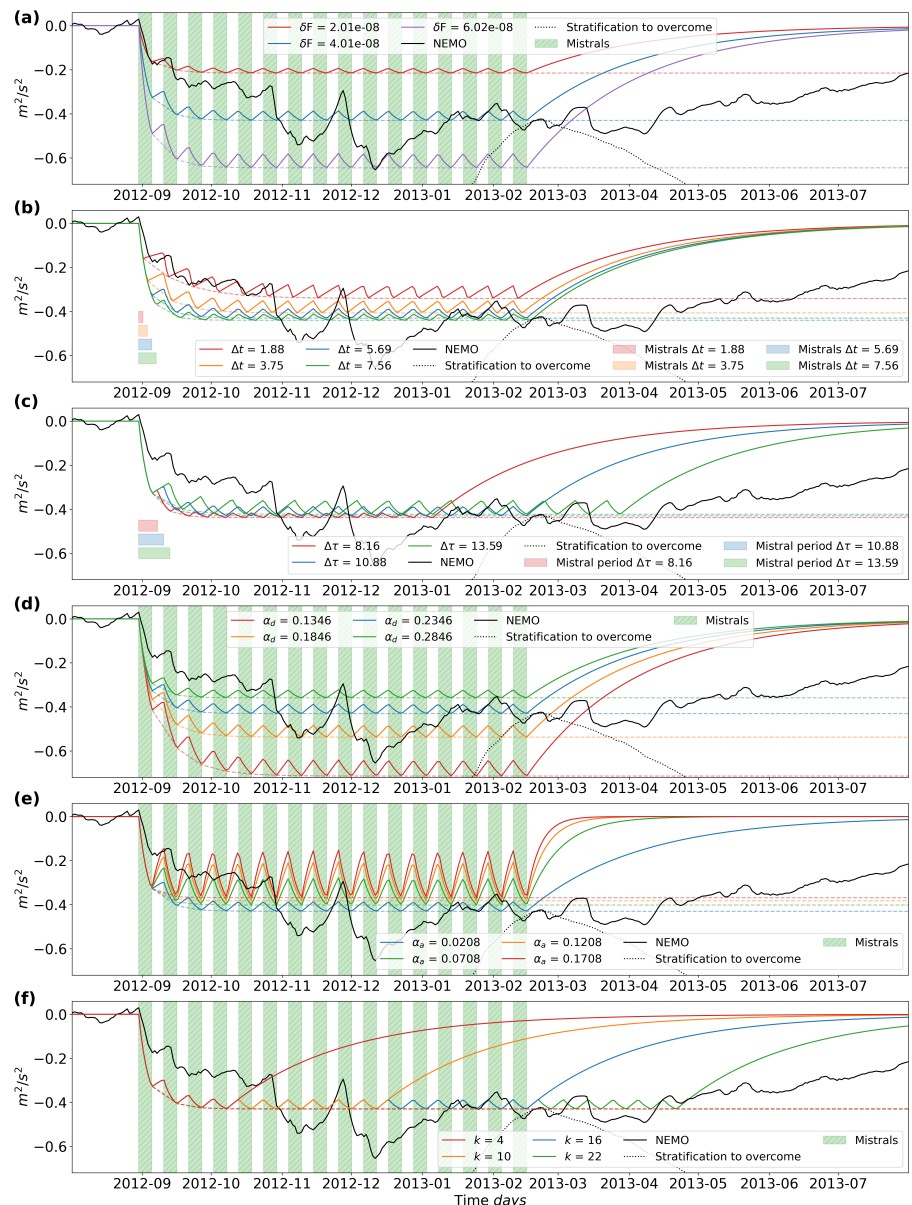

**Figure 12.** Equation (14) plotted with one variable varying in each plot with the other variables held constant at the mean value. (a) varies the strength of the Mistral, $\delta F$, (b) varies the duration, $\Delta t$, and (c) varies the period between events, $\Delta \tau$. (d) varies the restoration coefficient during the destratification phase, $\alpha_d$ and (e) varies the restoration coefficient for the restratification phase. (f) varies the number of events.

destratification phase of Eq. set (14) with respect to $k$, taking $t = \Delta \tau$, at the end of the phase, where the destratification equals the potential destratification:

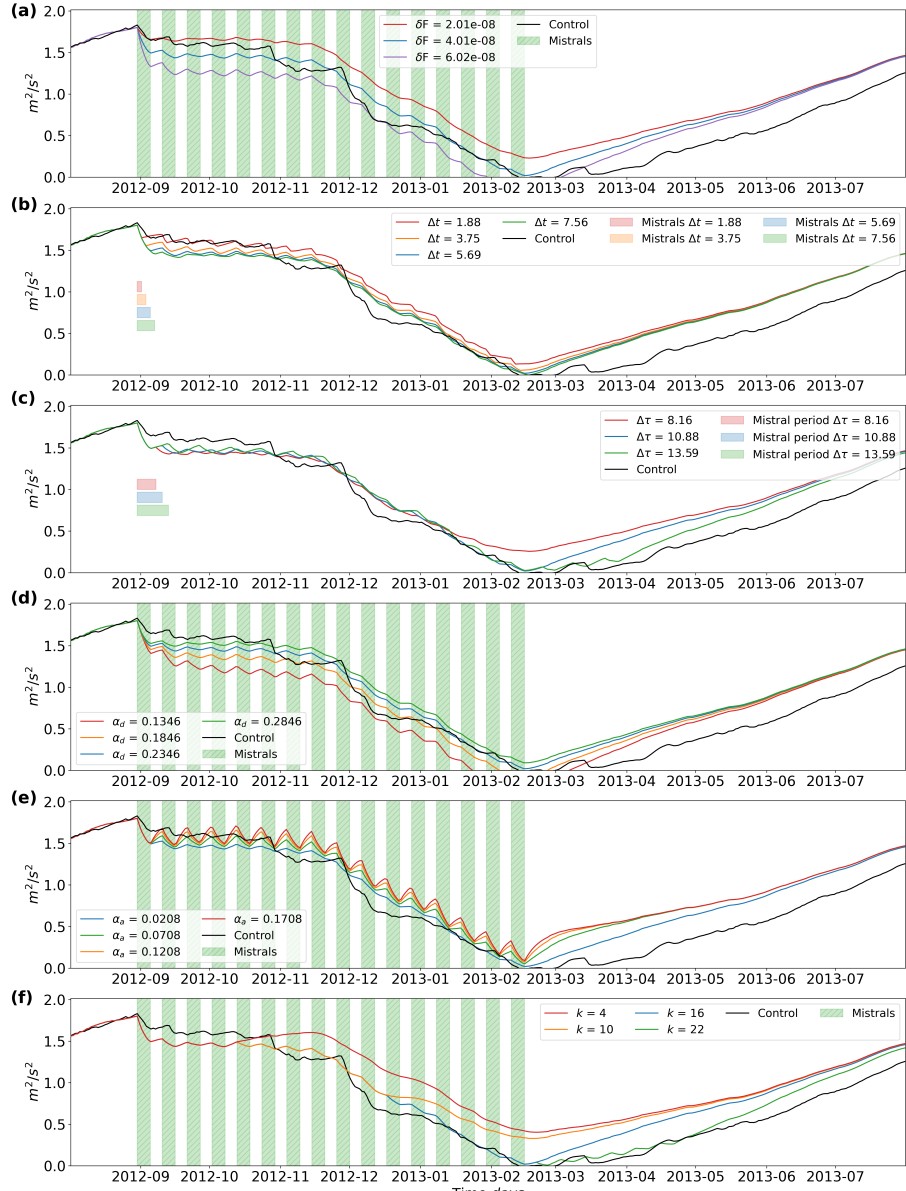

**Figure 13.** Same as Fig. 12, however, $SI_S$ is added to the results from Eq. (14).

$$\frac{\partial \delta SI_k(t=t_k+\Delta t)}{\partial k} = \frac{D^2}{2}\frac{\delta F}{\alpha_d}\frac{\left(e^{-\alpha_d \Delta t}-1\right)}{\left(1-e^{(\alpha_a-\alpha_d)\Delta t-\alpha_a \Delta \tau}\right)}\left(-e^{[(\alpha_a-\alpha_d)\Delta t-\alpha_a \Delta \tau]k}\right)\left((\alpha_a-\alpha_d)\Delta t-\alpha_a \Delta \tau\right) \tag{17}$$

Plugging in the mean values of $\Delta t$, $\Delta \tau$, and $\Delta F$, and taking $k = 16$, for the 16 events that occurred during the preconditioning phase, the above derivative equates a very small value of $-5.93 \times 10^{-11}$ $m^2/s^2$ per event. This confirms the small change in the potential destratification with increasing events. Taking $k$ to infinity and noting that $\alpha_d > \alpha_a$ results in the following:

$$\delta SI_\infty = \delta SI_\infty(t = t_k + \Delta t) = \frac{D^2}{2} \frac{\delta F}{\alpha_d} \left( e^{-\alpha_d \Delta t} - 1 \right) \left( \frac{1}{1 - e^{(\alpha_a - \alpha_d)\Delta t - \alpha_a \Delta \tau}} \right) \tag{18}$$

    We have an equation that describes the potential destratification, $\delta SI_\infty$, in terms of the Mistral attributes, independent of the
number of events. Differentiating by the different attributes (see App. A5 for the resulting analytical derivations) and plugging in the mean values where appropriate, we arrive at the resulting values: The derivative w.r.t. the strength of the Mistrals, $\partial \delta SI_\infty / \partial \Delta F$, equals a value of $-1.07 \times 10^7$ $m^2 day$, the derivative w.r.t. the duration, $\partial \delta SI_\infty / \partial \Delta t$, equals $-7.60 \times 10^{-3}$ $m^2/s^2 day$, and the derivative w.r.t. the period, $\partial \delta SI_\infty / \partial \Delta \tau$, equals $2.77 \times 10^{-3}$ $m^2/s^2 day$ (larger periods mean less frequent Mistral events, hence less destratification), respectively. Replacing $\delta SI$ with $\delta SI_\infty$ in Eq. (16), we can now multiply the partial
derivatives with the standard deviations to determine which attribute leads to the most potential destratification. The strength term is equal to $-1.28 \times 10^{-1}$ $m^2/s^2$, the duration term has a value of $-3.21 \times 10^{-2}$ $m^2/s^2$, and the period term has a value of $1.27 \times 10^{-2}$ $m^2/s^2$. With the strength term an order of magnitude larger than the other two terms, according to this simple model, the strength of the Mistral event is the most sensitive attribute when it comes to the effect of the Mistral on destratification, followed by its duration.

## 4.4  Simple model results

A complete and average Mistral destratification and restratification event according to Eq. (14) is given in Fig. 11 $c$, which took the average Mistral values from Table 1 and A1, and the restoring coefficients from App. A2. During the event, marked in green, the Mistral causes destratification. After the event, marked in blue, the ocean column restratifies until another event occurs (denoted by the dashed line). This is the same behavior we see in Fig. 8.

If we put together Eq. (13) with the duration and period information from Table 1, and Mistral strength information from Table A1, we can create a time series of $\delta SI$ to compare the integrated response of the simple model to the NEMO model results. This comparison is presented in Fig. 14. The simple model results resemble the NEMO simulation results quite well, which is expected as the fitted values for the restoring coefficients and the values for the Mistral event strengths are extracted from the NEMO model results. However, this means that a series of variable pulse like Mistral events can recreate with decent
accuracy the patterns that we see in the NEMO results for $\delta SI$. This essentially confirms that the Mistral events are the primary driving component of heat loss at the anomaly time scale leading to destratification.

## 5  Comparison with additional years

To understand the results of the 2013 deep convection year in a more generalized context, two additional years were simulated and analyzed in a similar fashion: the 1994 and 2005 winters (the June 1st, 1993 to May 31st, 1994 year, and the June 1st,

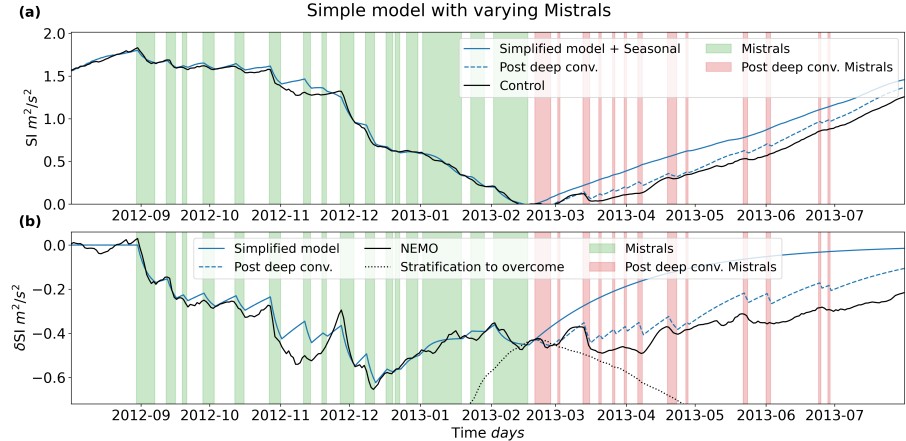

**Figure 14.** The combined effect of Eq. (13) for multiple Mistrals with the Mistral data from Table 1 and A1. (a) shows the calculated $\delta SI + SI_S$ response, while (b) is the calculated simple model $\delta SI$ versus the NEMO $\delta SI$ simulation results. Effects from Mistrals after deep convection are included with the dashed blue line and show that Mistrals after deep convection can retard the proceeding restratification during the restratification phase.

2004 to May 31st, 2005 year, respectively). The 2005 featured a deep convection event (Beuvier et al., 2012; Herrmann et al., 2010), whereas the 1994 winter did not (Somot et al., 2016). These years were chosen for the sake of having an additional deep convection year and year without deep convection, to see if there are any significant differences for non deep convection years and other deep convecting years. Simulations for the additional years were run in the same manner as the 2012 to 2013 year and with the same model and configuration. The Seasonal run similarly had its atmospheric forcing filtered with the same method

as in Sec. 2.2, with the Control run left unmodified. The only differences between these additional years' simulations and the 2012-2013 year simulations are the initial conditions, restoration data, and start dates of the simulations. For the additional years, the NEMO simulations were initialized with and restored to the MEDRYS reanalysis (Hamon et al., 2016). This was done as the initial conditions and restoration data for the 2012-2013 simulations were only available for that year. The starting time beginning in in June rather than July was an arbitrary decision and is not believed to significantly affect the results or

comparisons.

## 5.1    Stratification Index

As we are comparing separate years together, the time series simulation results were spatially averaged over a larger area for the additional years: from 42 to 42.5 $^\circ$ N and 4.25 to 5 $^\circ$ E. Fig. 15 and 16 show the $SI$ time series for the 1994 and 2005 winters, respectively. The increased spatial averaging reduces the extent at which the $SI$ destratifies, due to surrounding stratified water being averaged in, which makes the 2005 winter appear as though it's too stratified to deep convect ($0.22\ m^2/s^2$), even though

it's more destratified than the 1994 winter ($0.36\ m^2/s^2$). The MLD, however, clarifies that the 2005 winter experiences deep

convection in the simulation results and the 1994 winter does not (not shown), which is consistent with other findings (Beuvier et al., 2012; Herrmann et al., 2010; Somot et al., 2016).

Similar to the 2013 winter simulation set, the Seasonal run for the 2005 winter does not experience deep convection, again demonstrating the necessity to include forcing on both timescales for deep convection to occur. The 1994 winter reveals something conversely interesting: the $\delta SI$ is more negative for the 1994 winter, at -0.55 $m^2/s^2$, than for the 2013 winter at the time of minimum Control $SI$ (with deep convection in the latter but not the former), at -0.43 $m^2/s^2$. This means even a larger anomaly driven destratification is not able to overcome the residual stratification in a non convecting year, despite the fact that both the 1994 and 2005 winters each featured a lower maximum Control $SI$ than the 2013 winter: 1.83 $m^2/s^2$ for the 2013 winter and 1.73 and 1.79 $m^2/s^2$ for the 1994 and 2005 winters, respectively. This emphasizes the importance of the destratification caused by the seasonal forcing.

A note of interest for the 2005 and 2013 winters is the occurrence of the $\delta SI$ minimum. In the 2013 winter, the minimum occurs significantly before deep convection, in December. For the 2005 winter, the minimum occurs roughly about the time of deep convection (around the beginning of March; seen more clearly in the MLD; not shown), and also during a Mistral event, much like the 2013 winter. However, unlike the 2013 winter, the seasonal destratification is less active at the time of deep convection, whereas the anomaly destratification drops almost $-0.62 m^2/s^2$, most of it occurring during a larger Mistral event, to start deep convection. This suggests that the Mistral event occurring during this time triggers the deep convection event. While deep convection does not occur in the 1994 winter, the $\delta SI$ minimum is also at about the same time as the minimum in the Control $SI$, with a small Mistral event occurring at that date and with seasonal destratification remaining roughly constant. This suggests that if this year had further seasonal destratification or less initial destratification, the Mistral may have been the main trigger to deep convect as well, along with the larger Mistral event preceding it.

A note of interest for all three winters is the location of the Seasonal $SI$ minimum and the time of deep convection (or minimum of Control $SI$ for 1994). For the 2013 winter, the minimum is at roughly the time of deep convection, but is almost a month after in the 2005 winter. And for the 1994 winter, the minimum occurs before the Control minimum. This brings to question if the location of the Seasonal $SI$ minimum relative to the Control $SI$ minimum is important, and if so, how important is it in terms of deep convection occurring versus not.

## 5.2 Seasonal forcing

The seasonal sea surface fluxes for both years resemble the fluxes of the 2013 winter (see Fig. 17). The solar radiation component drives the main shape of the $SI$ time series, with the major component contributing to the asymmetry being the latent heat flux, followed by the sensible heat flux. However, in the 2005 winter, the latent heat flux has a larger heat loss value than the 1994 year, reaching over 300 $W/m^2$ versus under 200 $W/m^2$ in the latter, driving the net heat flux more negative and causing more destratification according to the simple model, resulting in deep convection encompassing a few days on either side of the beginning of March.

The simple model for the seasonal $SI$ is fairly accurate for the 1994 winter, similar to the 2013 winter, but is not quite as accurate for the 2005 winter (see Fig. 17). For the 2005 winter, the simple model deviates during the deepest part of the

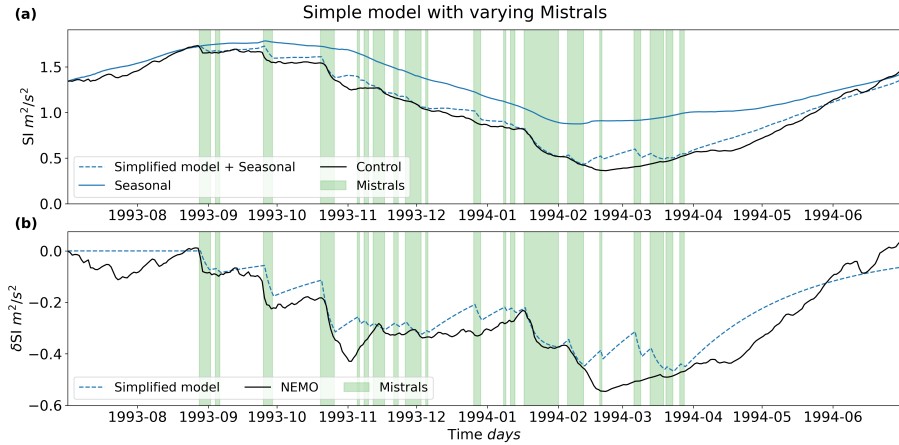

**Figure 15.** The Stratification Index for the 1994 winter for both the Control and Seasonal runs are in subplot (a), spatially averaged over the area of 42 to 42.5 ° N and 4.25 to 5 ° E. The simplified model anomaly solution added to the Seasonal $SI$ is denoted by the dashed line. Subplot (b) shows the NEMO determined $\delta SI$ and the $\delta SI$ calculated from the anomaly solution of the simple model.

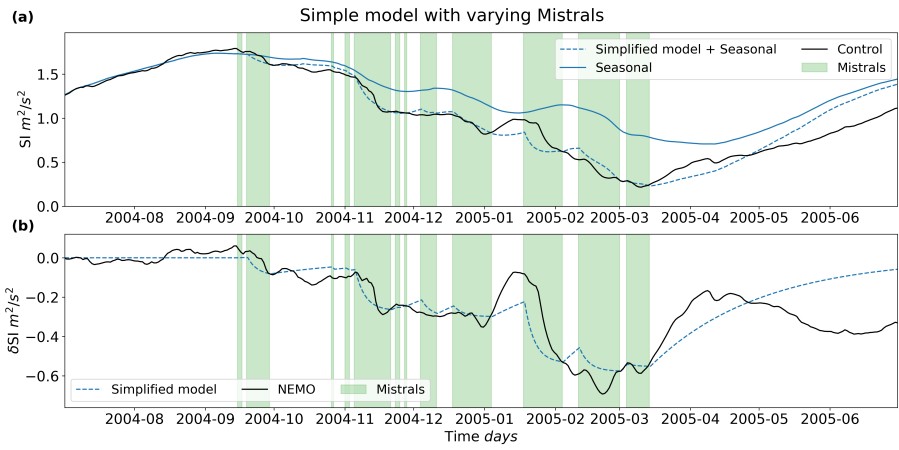

**Figure 16.** Same as Fig. 15 for the 2005 winter.

winter. This could be due to more advective behavior captured by the destratification with the larger spatial averaging, which is neglected in the seasonal component of the simple model.

### 5.3 Anomaly forcing

The simple model for the anomaly scale was calculated for both the 1994 and 2005 winters, following the same steps in Sec.
4.3. The value of the restoration coefficients, $\alpha_d$ and $\alpha_a$, were carried over from the 2013 winter analysis, with the Mistral dates determined through the same process outlined in Sec. 2.3 and were manually adjusted to fit visual data (again with the

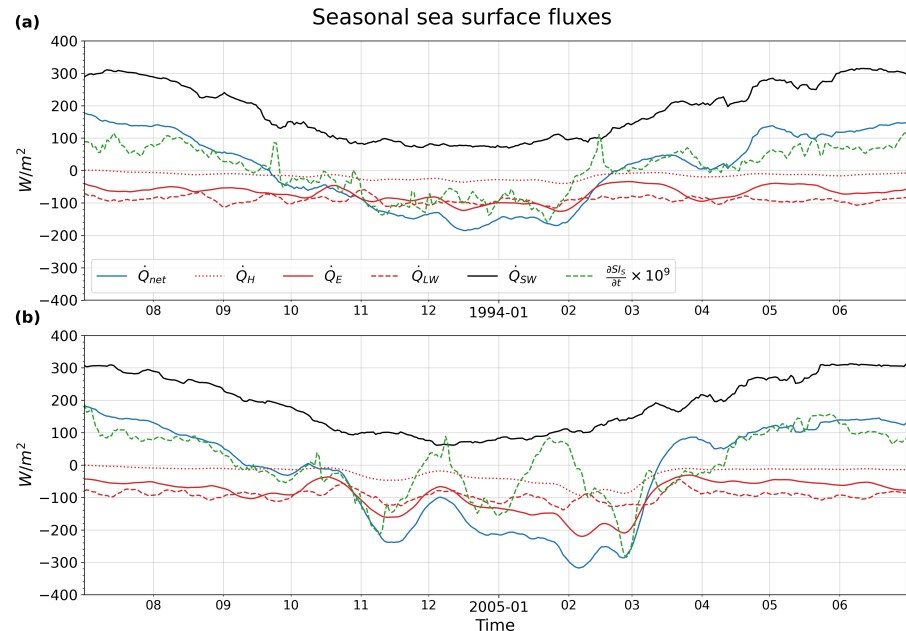

**Figure 17.** The smoothed seasonal sea surface fluxes spatially averaged over the 42 to 42.5 ° N and 4.25 to 5 ° E area, with the green dashed line denoting the estimated derivative w.r.t. time of the seasonal $SI$ from the NEMO results, multiplied by $10^9$ $m^4/Js^2$. A negative value means heat is leaving the ocean.

same method described in Sec. 2.3). The Mistral strengths of the events during the preconditioning phase were determined through the same process as in Sec. 4.3. The Mistral dates and strengths are presented in the appendix (App. B). The results are shown in Fig. 15 and 16, and the simple model follows quite closely to the NEMO simulation results, only deviating majorly at

extreme peaks and troughs, despite utilizing the restoration coefficients from the 2013 winter. This reinforces the importance of the Mistral as a dominating factor for destratification on the anomaly time scale.

The different components of Mistral (strength, $\delta F$, duration, $\Delta t$, and period, $\Delta \tau$) are separated in the same manner as for the 2013 winter, to determine the main factor of the Mistral leading to destratification, according to these additional years. For the 1994 winter, the contribution due to strength, duration, and period equal: $-8.71 \times 10^{-2}$, $-6.39 \times 10^{-2}$, and $5.83 \times 10^{-2}$

$m^2/s^2$ (recall that a larger period means less frequent Mistral events). For the 2005 winter, the contribution due to strength, duration, and period equal: $-1.51 \times 10^{-1}$, $-5.23 \times 10^{-2}$, and $4.89 \times 10^{-2}$ $m^2/s^2$. The 2005 winter results have the same order of magnitude as the 2013 winter results, with all three winters having the same order of importance for the Mistral attributes: first strength, then duration, followed by the length of the period. Only for the 1994 winter was the strength term found not to be as dominant as in the other years, with the term having the same order of magnitude as the other terms. However, as

the order of importance was still the same for all three years, this aids the conclusion that, in general, the strength term of the Mistral is its most important factor driving destratification.

## 6 Conclusions

The 2012-2013 deep convection year (2013 winter) in the Gulf of Lion was investigated to determine the effect the Mistral winds have on deep convection. Two NEMO ocean simulations were run, one forced with unmodified WRF/ORCHIDEE atmospheric forcing (Control) and one forced with atmospheric fields filtered to remove the Mistral signature (Seasonal). Separating the atmospheric forcing into the long-term and anomaly timescales revealed that the Mistral winds do not act alone to destabilize the northwestern Mediterranean Sea. Both the seasonal atmospheric change, reflected in the long-term timescales, and the Mistral winds, reflected in the anomaly timescales, combine to destabilize and destratify the water columns in the GOL in roughly equal amounts (favoring the seasonal change).

When the NEMO simulation results were probed further by developing a simple model, the simple model conveyed the underlying drivers of the long-term, or seasonal timescale. The evolution of the seasonal Stratification Index is proportional to the net heat flux leaving the ocean. As the net heat flux follows the shape of the incoming solar radiation, the maximum and minimum values for the seasonal Stratification Index occur around Sept. 21st and March 21st, respectively, or the fall and spring equinoxes. Shifted negative by the latent, sensible, and longwave radiation heat fluxes, the net heat flux allows for a seasonal cycle of destratification during the winter and restratification during the summer. If any of the three negative shifting components are unable to cool the ocean surface enough, deep convection may fail to appear, unless the contribution of the Mistral winds is able to compensate.

The simple model results go on to confirm the hypothesis that the Mistral acts on the anomaly timescale to destratify the water column, and is the primary driver in this timescale. These results further conveyed that additional Mistral events need to be stronger in terms of heat transfer than previous events to create further destratification. Otherwise, no destratification, or even restratification, occurs. The simple model then goes on to reveal, after some additional derivation, that the most important part of a Mistral event is its strength, in regards to potential destratification. Changing the duration or frequency has an effect, but this effect is on a order of magnitude smaller than changing the Mistral strength.

Two additional years were also studied with the same method of running a Control simulation and a filtered atmospheric forcing Seasonal simulation: the year of 1993-1994 (1994 winter) and the year of 2004-2005 (2005 winter). These years were then also studied with the simple model framework. The 2005 winter featured a deep convection event like the 2013 winter, but the 1994 winter did not, allowing for the comparison between deep convecting and non deep convecting years. The conclusions determined in the 2013 winter are largely supported by the results of the two additional years. The seasonal change in $SI$ accounts for a larger part of the destratification, while the 2005 winter still required destratification from the anomaly scale to deep convect. The solar radiation component of the seasonal forcing was also found to be the component giving the cyclical structure to the Stratification Index, with the latent and sensible heat fluxes creating the asymmetry. On the anomaly scale forcing, the Mistral strength was again found to be the dominating component leading to destratification, although its magnitude is slightly less pronounced in the 1994 winter.

However, some of the NEMO simulation results bring further questions. Sec. 3 noted that the seasonal change in stratification brought the preconditioned water column in the GOL to the point of deep convection simultaneously with a Mistral

event, for the winter of 2012-2013, as the Mistral induced preconditioning had already passed its minimum destratification beforehand, with both the seasonal change and a Mistral event acting to destratify at the moment of deep convection. In the 2005 winter, the maximum seasonal destratification had not yet occurred during the time of deep convection, but the Mistral induced destratification brought the stratification down to point of deep convection, triggering it with a Mistral event. Despite

deep convection not occurring, a similar structure appears in the $SI$ and $\delta SI$ for the winter of 1994. Additionally, the date of the Seasonal $SI$ minimum was different relative to the date of deep convection (Control $SI$ minimum for the 1994 winter), for all three winters.

This brings up two questions. The first is whether or not the Mistral truly triggers deep convection for all deep convection events, or if the change in the seasonal destratification at the time of deep convection is a more prominent factor. The second is,

what is the importance of the location of the Seasonal $SI$ minimum and does it make a difference in regard to the possibility of deep convection. Is it possible for the Mistral induced destratification to cause deep convection after the minimum has passed, despite being hindered by the restratifying Seasonal $SI$.

Another question brought around by the simulation results is what is the effect of the maximal stratification at the beginning of the preconditioning phase on the ocean's ability to experience deep convection for a given year? The maximal stratification

must be overcome by the Mistral and seasonal forcing for deep convection to occur, with the ability of the forcing to do so varying per year. For our results, the 2013 winter had a larger maximum Control $SI$ than the other two winters investigated and still deep convected. Similarly, the 2005 winter had a larger maximum stratification than the winter of 1994: 1.79 vs. 1.73 $m^2/s^2$. However, all three winters had a maximal stratification within $0.1 \ m^2/s^2$ of each other, which is only about 6% of the maximum.

For example, for the 2004-2005 winter the atmospheric forcing was more than enough to overcome the initial stratification and a milder winter would have lead to deep convection as well, according to other works (Grignon et al., 2010). Somot et al. (2016) investigated initial stratification and over a longer time period than Grignon et al. (2010) (1995-2005 vs. 1980-2013), however they calculated the Stratification Index at the beginning of December rather than at the beginning of September, where the maximum stratification of 1.83 $m^2/s^2$ occurs for the 2012-2013 winter. By December, the $SI$ already dropped to

1 $m^2/s^2$, which means almost half of the destratification has already occurred, with their calculation missing about half of the preconditioning phase. The same is true when looking at the 1994 and 2005 winters. A maximum $SI$ calculated near the beginning of September may be more representative of the water columns ability to deep convect and needs to be investigated. A related question is how does the accumulation (or reduction) of stratification transferred to the proceeding year affect deep convection in the following years? For the winter of 2005, where very few of the preceding 15 years deep convected due to

milder winters, the result was warmer and saltier WDMW production (Herrmann et al., 2010).

These questions are outside the scope of the current article, as they rely on investigating a large number of multiple years to evaluate the inter-annual variability of the atmospheric forcing. The 2013 winter featured an above average year in terms of destratification, leading to deep convection, while multiple years in the 1990s saw minimal MLD growth (including the 1994 winter; Somot et al. (2016) and references therein). This may have been due to an above average number of Mistral (stormy)

550 days, as suggested by Somot et al. (2016), which coincides with our results of 36%, 61%, and 54% of the preconditioning days

having a Mistral event for the winters of 1994, 2005, and 2013, respectively. But it may have also been due to a larger than average contribution from the seasonal forcing, as the seasonal $SI$ saw more destratification than the anomaly stratification index, $\delta SI$, which isn't clearly discernible with just three winters.

We believe the approach of separating the atmospheric forcing into the seasonal and anomaly components will reveal more answers to these questions over larger set of multiple years and we are preparing additional works to address them. We hope these works will provide us with more information on how the Gulf of Lion deep convection system will evolve in the future.

## Appendix A: Equations

### A1  Simple Stratification Index model

The purpose of this simple model is to separate the seasonal scale atmospheric forcing from the anomaly scale forcing. We start the derivation with the energy equation for incompressible flow (White, 2011):

$$\rho c_p \frac{DT}{Dt} = \frac{D\mathbf{q}}{Dt} \tag{A1}$$

where $\rho$ is density, $c_p$ is the specific heat of the fluid with constant pressure, $T$ is temperature, $t$ is time, and $\mathbf{q}$ is energy per volume from heat. $\frac{D}{Dt}$ is the material derivative.

In this model we're assuming the heat transfer term is equal to the heat removed by the atmosphere:

$$\frac{D\mathbf{q}}{Dt} = -\mathbf{q_a} \tag{A2}$$

where $\mathbf{q_a}$ is the volumetric heat forcing from the atmosphere. This leaves us with the following equation:

$$\frac{DT}{Dt} = -\frac{\mathbf{q_a}}{\rho c_p} \tag{A3}$$

The Brunt-Väisälä frequency is defined as:

$$N^2 = \frac{\partial b}{\partial z} = -\frac{g}{\rho_0}\frac{\partial \rho}{\partial z} \tag{A4}$$

Assuming a fluid whose density varies negatively proportionally to the temperature, $\rho = -\beta T$, which is an acceptable approximation as the density only varies only a few tenths of a $kg/m^3$ and temperature only varies about 10 degrees Celsius, we can describe $N^2$ in terms of temperature:

$$N^2 = \frac{\partial b}{\partial z} = -\frac{g}{T_0}\frac{\partial T}{\partial z} \tag{A5}$$

Introducing buoyancy as $b = -\frac{g}{T_0}T$, we can rearrange the energy equation into terms of buoyancy:

$$575 \quad \frac{g}{T_0}\left[\frac{DT}{Dt} = -\frac{\mathbf{q_a}}{\rho c_p}\right] \Rightarrow \frac{Db}{Dt} = -\frac{\mathbf{q_a}g}{\rho c_p T_0} \tag{A6}$$

If we then differentiate by $\frac{\partial}{\partial z}$, we can reorganize the equation in terms of the Brunt-Väisälä frequency, $N^2$:

$$\frac{\partial}{\partial z}\left(\frac{Db}{Dt} = -\frac{\mathbf{q_a}g}{\rho c_p T_0}\right) \Rightarrow \frac{DN^2}{Dt} = -\frac{\partial}{\partial z}\left(\frac{\mathbf{q_a}g}{\rho c_p T_0}\right) \tag{A7}$$

By renaming the atmospheric forcing term on the right hand side of Eq. (A7) to $F(t)$, we can make this equation easier to follow:

$$580 \quad F(t) = \frac{\partial}{\partial z}\left(\frac{\mathbf{q_a}g}{\rho c_p T_0}\right) \tag{A8}$$

This brings us to:

$$\frac{DN^2}{Dt} = -F(t) \tag{A9}$$

The main assumption we make is that the ocean column is a linear system and responds to the large and short/anomaly time scale atmospheric forcing independently:

$$585 \quad \begin{aligned} N^2 &= N_S^2 + \delta N^2 \\ F(t) &= \delta F(t) + F_S(t) \end{aligned} \tag{A10}$$

which describes the response of $N^2$ on the anomaly timescale, $\delta N^2$:

$$\frac{D\delta N^2}{Dt} = -\delta F(t) \tag{A11}$$

and the response of $N^2$ on the seasonal timescale, $N_S^2$:

$$\frac{DN_S^2}{Dt} = -F_S(t) \tag{A12}$$

For the seasonal response, we further make the assumption that $N_S^2$ negligibly depends on the $x, y$, and $z$ coordinate directions:

$$\frac{dN_S^2}{dt} = -F_S(t) \tag{A13}$$

If we want to connect the overall Brunt-Väisälä frequency, $N^2$, to the seasonal one, $N_S^2$, we can formulate a restoring term, $R$, in terms of $T$, or in terms of $N^2$ following the steps mentioned above:

$$R = \frac{\partial}{\partial z}\left(\frac{g}{T_0}\alpha\rho c_p(T - T_S)\right) \Rightarrow \alpha(N^2 - N_S^2) \tag{A14}$$

Or, with $\delta N^2 = N^2 - N_S^2$:

$$R = \alpha\delta N^2 \tag{A15}$$

Where $\alpha$ is the restoring term coefficient. Separating the material derivative into its time and advective components for Eq. (A11):

$$\frac{\partial\delta N^2}{\partial t} + \boldsymbol{V}\cdot\nabla(\delta N^2) = -\delta F(t) \tag{A16}$$

we will replace the advective component, $\boldsymbol{V}\cdot\nabla(\delta N^2)$, with $R$, which essentially swallows the advective operation into the restoring coefficient, $\alpha$. This results in the partial differential equation that we will study further:

$$\frac{\partial\delta N^2}{\partial t} + \alpha\delta N^2 = -\delta F(t) \tag{A17}$$

### A1.1 Solution for seasonal $SI$

To solve the response of $N^2$ for the seasonal timescale, given by Eq. (A13), we will assume $N_S^2$ is vertically homogeneous, giving us the stratification index response, or Eq. (9), through the use of Eq. (6). We can then separate back out $F_S(t)$ into its components:

$$F_S(t) = \frac{g}{\rho c_p T_0}\frac{\partial\boldsymbol{q_{a,S}}}{\partial z} \tag{A18}$$

Dividing $\boldsymbol{q_{a,S}}$ by $D$ gives us the atmospheric cooling in terms of a surface flux, $-Q_{net,S}$. If we plug this relationship back into Eq. (9), we get:

$$\frac{\mathrm{d}SI_S}{\mathrm{d}t} = \frac{D}{2}\frac{g}{\rho c_p T_0}\frac{\partial Q_{net,S}}{\partial z} \tag{A19}$$

Integrating this equation by $z$ gives us:

$$\int\limits_0^D \frac{\mathrm{d}SI_S}{\mathrm{d}t}\partial z = \frac{D}{2}\frac{g}{\rho c_p T_0}\int\limits_0^D \frac{\partial Q_{net,S}}{\partial z}\partial z$$

$$\frac{\mathrm{d}SI_S}{\mathrm{d}t}D = \frac{D}{2}\frac{g}{\rho c_p T_0}Q_{net,S} \tag{A20}$$

$$\frac{\mathrm{d}SI_S}{\mathrm{d}t} = \frac{g}{2\rho c_p T_0}Q_{net,S}$$

And, therefore, we have $SI_S$ expressed in terms of $Q_{net,S}$.

## A1.2 Solution with Mistral forcing function

Focusing on just the anomaly time scale, we will assume the Mistral is the primary source of forcing. To model the atmospheric cooling of the Mistral, we will model the forcing function, $\delta F(t)$, as a series of $k$ pulse functions, of magnitude $\delta F_k$, over a duration of $\Delta t_k$, and with a period of $\Delta \tau_k$, visualized in Fig. 10.

To solve the Brunt-Väisälä frequency response with the Mistral pulse forcing function, we solve Eq. (A17) in a piecewise manner, with a solution for each section of the pulse function. We will also make the assumption that for each portion of the Mistral event, during and after, the advective components, hence $\alpha$, remain constant with respect to time. This leads to $\alpha_d$ and $\alpha_a$ representing the advective components during and after an event, respectively. During a Mistral event, $[t_k, t_k + \Delta t_k)$, we get:

$$\frac{\partial \delta N^2}{\partial t} + \alpha_d(\delta N^2) = -\delta F(t)$$

$$\delta N_k^2(t) = -\frac{\delta F_k}{\alpha_d} + c_0 e^{-\alpha_d t}$$

$$\delta N_k^2(t_k) = -\frac{\delta F_k}{\alpha_d} + c_0 e^{-\alpha_d t_k} = \delta N_{k-1}^2(t_k) \tag{A21}$$

$$c_0 = \left[\delta N_{k-1}^2(t_k) + \frac{\delta F_k}{\alpha_d}\right]e^{\alpha_d t_k}$$

With the result:

$$\delta N_k^2(t) = \left[\delta N_{k-1}^2(t_k) + \frac{\delta F_k}{\alpha_d}\left(1 - e^{\alpha_d(t-t_k)}\right)\right]e^{-\alpha_d(t-t_k)} \tag{A22}$$

After the event, $[t_k + \Delta t_k, t_{k+1})$:

$$\frac{\partial \delta N^2}{\partial t} + \alpha_a (\delta N^2) = 0$$

$$\delta N_k^2(t) = c_1 e^{-\alpha_a t}$$

$$\delta N_k^2(t_k + \Delta t_k) = c_1 e^{-\alpha_a (t_k + \Delta t_k)} \tag{A23}$$

$$= \left[ \delta N_{k-1}^2(t_k) + \frac{\delta F_k}{\alpha_d} \left( 1 - e^{\alpha_d \Delta t_k} \right) \right] e^{-\alpha_d \Delta t_k}$$

$$c_1 = \left[ \delta N_{k-1}^2(t_k) + \frac{\delta F_k}{\alpha_d} \left( 1 - e^{\alpha_d \Delta t_k} \right) \right] e^{(\alpha_a - \alpha_d)\Delta t_k - \alpha_a t_k}$$

With the result:

$$\delta N_k^2(t) = \left[ \delta N_{k-1}^2(t_k) + \frac{\delta F_k}{\alpha_d} \left( 1 - e^{\alpha_d \Delta t_k} \right) \right] e^{(\alpha_a - \alpha_d)\Delta t_k - \alpha_a (t - t_k)} \tag{A24}$$

Or, to have the results more succinctly put:

$$\delta N_k^2(t) = \begin{cases} \left[ \delta N_{k-1}^2(t_k) + \frac{\delta F_k}{\alpha_d} \left( 1 - e^{\alpha_d (t - t_k)} \right) \right] e^{-\alpha_d (t - t_k)} & [t_k, t_k + \Delta t_k) \\ \left[ \delta N_{k-1}^2(t_k) + \frac{\delta F_k}{\alpha_d} \left( 1 - e^{\alpha_d \Delta t_k} \right) \right] e^{(\alpha_a - \alpha_d)\Delta t_k - \alpha_a (t - t_k)} & [t_k + \Delta t_k, t_k + \Delta \tau_k) \end{cases} \tag{A25}$$

### A1.3 $\delta N_{k-1}^2$ initial condition

$\delta N_{k-1}^2(t_k)$ is a recursive initial condition, as its initial condition is the event before it, and so on:

$$\delta N_{k-1}^2(t_k) = \left[ \delta N_{k-2}^2(t_{k-1}) + \frac{\delta F_{k-1}}{\alpha_d} (1 - e^{\alpha_d \Delta t_{k-1}}) \right] e^{(\alpha_a - \alpha_d)\Delta t_{k-1} - \alpha_a \Delta \tau_{k-1}}$$

$$\delta N_{k-2}^2(t_{k-1}) = \left[ \delta N_{k-3}^2(t_{k-2}) + \frac{\delta F_{k-2}}{\alpha_d} (1 - e^{\alpha_d \Delta t_{k-2}}) \right] e^{(\alpha_a - \alpha_d)\Delta t_{k-2} - \alpha_a \Delta \tau_{k-2}} \tag{A26}$$

Therefore, $\delta N_{k-1}^2(t_k)$ can be simplified in expression by combining the initial conditions:

$$\delta N_{k-1}^2(t_k) = \delta N_{k-m}^2(t_{k-(m-1)}) e^{(\alpha_a - \alpha_d)\sum_{i=1}^{m-1} \Delta t_{k-i}} e^{-\alpha_a \sum_{i=1}^{m-1} \Delta \tau_{k-i}}$$

$$+ \sum_{j=1}^{m-1} \frac{\delta F_{k_j}}{\alpha_d} \left( 1 - e^{\alpha_d \Delta t_{k-j}} \right) e^{(\alpha_a - \alpha_d)\sum_{i=1}^{j} \Delta t_{k-i}} e^{-\alpha_a \sum_{i=1}^{j} \Delta \tau_{k-i}} \tag{A27}$$

If $m = k$ and $\delta N_0^2 = 0$:

$$\delta N_{k-1}^2(t_k) = \sum_{j=1}^{k-1} \frac{\delta F_{k_j}}{\alpha_d} \left( 1 - e^{\alpha_d \Delta t_{k-j}} \right) e^{(\alpha_a - \alpha_d)\sum_{i=1}^{j} \Delta t_{k-i}} e^{-\alpha_a \sum_{i=1}^{j} \Delta \tau_{k-i}} \tag{A28}$$

Assuming $\delta F_k = \delta F$, $\Delta t_k = \Delta t$, and $\Delta \tau_k = \Delta \tau$ for all $k$, or a periodic pulse function, then $\delta N^2_{k-1}$ can be expressed as:

$$\delta N^2_{k-1}(t_k) = \frac{\delta F}{\alpha_d}\left(1 - e^{\alpha_d \Delta t}\right)\sum_{j=1}^{k-1} e^{[(\alpha_a - \alpha_d)\Delta t - \alpha_a \Delta \tau]j} \tag{A29}$$

Taking the sum of a finite geometric series:

$$\sum_{n=0}^{m-1} r^n = \left(\frac{1 - r^m}{1 - r}\right)$$
$$\sum_{n=0}^{m-1} r^n = \sum_{n=1}^{m-1} r^n + 1 \tag{A30}$$
$$\sum_{n=1}^{m-1} r^n = \sum_{n=0}^{m-1} r^n - 1$$

where $r \neq 1$. If $r = e^{(\alpha_a - \alpha_d)\Delta t - \alpha_a \Delta \tau}$:

$$\sum_{j=1}^{k-1} e^{[(\alpha_a - \alpha_d)\Delta t - \alpha_a \Delta \tau]j} = \left(\frac{1 - e^{[(\alpha_a - \alpha_d)\Delta t - \alpha_a \Delta \tau]k}}{1 - e^{[(\alpha_a - \alpha_d)\Delta t - \alpha_a \Delta \tau]}} - 1\right) \tag{A31}$$

we then we get:

$$\delta N^2_{k-1}(t_k) = \frac{\delta F}{\alpha_d}\left(1 - e^{\alpha_d \Delta t}\right)\left(\frac{1 - e^{[(\alpha_a - \alpha_d)\Delta t - \alpha_a \Delta \tau]k}}{1 - e^{[(\alpha_a - \alpha_d)\Delta t - \alpha_a \Delta \tau]}} - 1\right) \tag{A32}$$

where $(\alpha_a - \alpha_d)\Delta t - \alpha_a \Delta \tau \neq 0$. Plugging Eq. (A32) into Eq. (A22) and (A24) results in the equation for the response of the Brunt-Väisälä frequency forced by a periodic pulse function:

$$\delta N^2(t) = \begin{cases} \frac{\delta F}{\alpha_d}\left[\left(1 - e^{\alpha_d \Delta t}\right)\left(\frac{1 - e^{[(\alpha_a - \alpha_d)\Delta t - \alpha_a \Delta \tau]k}}{1 - e^{[(\alpha_a - \alpha_d)\Delta t - \alpha_a \Delta \tau]}} - 1\right) + \left(1 - e^{\alpha_d(t - t_k)}\right)\right]e^{-\alpha_d(t - t_k)} & [t_k, t_k + \Delta t) \\ \frac{\delta F}{\alpha_d}\left[\left(1 - e^{\alpha_d \Delta t}\right)\left(\frac{1 - e^{[(\alpha_a - \alpha_d)\Delta t - \alpha_a \Delta \tau]k}}{1 - e^{[(\alpha_a - \alpha_d)\Delta t - \alpha_a \Delta \tau]}} - 1\right) + \left(1 - e^{\alpha_d \Delta t}\right)\right]e^{(\alpha_a - \alpha_d)\Delta t - \alpha_a(t - t_k)} & [t_k + \Delta t, t_k + \Delta \tau) \end{cases} \tag{A33}$$

For the anomaly response, Eq. (A25) and (A33), assuming a vertically homogeneous $\delta N^2$ leads to the stratification index through Eq. (6), leads us to $\delta SI$ being expressed as:

$$\delta SI_k(t) = \begin{cases} \left[\delta SI^2_{k-1}(t_k) + \frac{D^2}{2}\frac{\delta F_k}{\alpha_d}\left(1 - e^{\alpha_d(t - t_k)}\right)\right]e^{-\alpha_d(t - t_k)} & [t_k, t_k + \Delta t_k) \\ \left[\delta SI^2_{k-1}(t_k) + \frac{D^2}{2}\frac{\delta F_k}{\alpha_d}\left(1 - e^{\alpha_d \Delta t_k}\right)\right]e^{(\alpha_a - \alpha_d)\Delta t_k - \alpha_a(t - t_k)} & [t_k + \Delta t_k, t_k + \Delta \tau_k) \end{cases} \tag{A34}$$

And:

$$\delta SI_k(t) = \begin{cases} \frac{D^2}{2} \frac{\delta F}{\alpha_d} \left[ \left(1 - e^{\alpha_d \Delta t}\right) \left( \frac{1 - e^{[(\alpha_a - \alpha_d)\Delta t - \alpha_a \Delta \tau]k}}{1 - e^{[(\alpha_a - \alpha_d)\Delta t - \alpha_a \Delta \tau]}} - 1 \right) + \left(1 - e^{\alpha_d(t - t_k)}\right) \right] e^{-\alpha_d(t - t_k)} & [t_k, t_k + \Delta t) \\[3mm] \frac{D^2}{2} \frac{\delta F}{\alpha_d} \left[ \left(1 - e^{\alpha_d \Delta t}\right) \left( \frac{1 - e^{[(\alpha_a - \alpha_d)\Delta t - \alpha_a \Delta \tau]k}}{1 - e^{[(\alpha_a - \alpha_d)\Delta t - \alpha_a \Delta \tau]}} - 1 \right) + \left(1 - e^{\alpha_d \Delta t}\right) \right] e^{(\alpha_a - \alpha_d)\Delta t - \alpha_a(t - t_k)} & [t_k + \Delta t, t_k + \Delta \tau) \end{cases}$$

$$(A35)$$

for the period pulse function case.

## A2  Restoring coefficients, $\alpha_d$ and $\alpha_a$

The restoration coefficients, $\alpha_d$ and $\alpha_a$, can be solved for the separate phases of a Mistral event in Eq. (A34) by normalizing the equations during their respective phases. These normalized equations are then fitted to selected, ideal Mistral destratification and restratification cases that are highlighted in Table 1 and Fig. 8 to retrieve the values of the restoration coefficients.

### A2.1  Restoration coefficient $\alpha_d$, during a Mistral

To solve for $\alpha_d$, we normalize Eq. (A34) for during a Mistral event, $[t_k, t_k + \Delta t_k)$, with $\delta SI$ given as:

$$\delta SI_k(t) = \left[ \delta SI_{k-1}^2(t_k) + \frac{D^2}{2} \frac{\delta F_k}{\alpha_d} \left(1 - e^{\alpha_d(t - t_k)}\right) \right] e^{-\alpha_d(t - t_k)} \tag{A36}$$

We first reference the time, $t$, to the starting time of event $k$ as $t' = t - t_k$, giving us:

$$\delta SI_k(t') = \left[ \delta SI_{k-1}^2(t_k) + \frac{D^2}{2} \frac{\delta F_k}{\alpha} \left(1 - e^{\alpha_d t'}\right) \right] e^{-\alpha_d t'} \tag{A37}$$

Next, we normalize $\delta SI_k$ to the value of zero at $t' = 0$, resulting in $\delta SI_{k,NI}$:

$$\begin{aligned} \delta SI_{k,NI}(t') = \delta SI_k(t') - \delta SI_k(t' = 0) &= \left[ \delta SI_{k-1}(t_k) + \frac{D^2}{2} \frac{\delta F_k}{\alpha_d} \left(1 - e^{\alpha_d t'}\right) \right] e^{-\alpha_d t'} - \delta SI_{k-1}(t_k) \\ &= \left[ \delta SI_{k-1}(t_k) + \frac{D^2}{2} \frac{\delta F_k}{\alpha_d} \right] \left( e^{-\alpha_d t'} - 1 \right) \end{aligned} \tag{A38}$$

Then the height or magnitude of destratification for each event is normalized to 1, resulting in $\delta SI_{k,NH}$:

$$\delta SI_{k,NH}(t') = \frac{\delta SI_{k,NI}(t')}{\text{extremum}(\delta SI_{k,NI}(t'))} \tag{A39}$$

The extremum value for $\delta SI_{k,NI}(t')$ is when $t' = \Delta t_k$, or at the end of the Mistral event. This simplifies $\delta SI_{k,NH}$ to:

$$\delta SI_{k,NH}(t') = \frac{\left[ \delta SI_{k-1}(t_k) + \frac{D^2}{2} \frac{\delta F_k}{\alpha_d} \right] \left( e^{-\alpha_d t'} - 1 \right)}{\left[ \delta SI_{k-1}(t_k) + \frac{D^2}{2} \frac{\delta F_k}{\alpha_d} \right] \left( e^{-\alpha_d \Delta t_k} - 1 \right)} = \frac{\left( e^{-\alpha_d t'} - 1 \right)}{\left( e^{-\alpha_d \Delta t_k} - 1 \right)} \tag{A40}$$

Then, to normalize the length of the event duration from 0 to 1, we divide $t'$ by the event length, $\Delta t_k$, which results in $t''$:

$$t'' = \frac{t'}{\Delta t_k} \Rightarrow t' = t'' \Delta t_k \tag{A41}$$

Plugging $t''$ into $\delta SI_{k,NH}(t')$ returns $\delta SI_{k,NT}$:

$$\delta SI_{k,NT}(t'') = \frac{e^{-\alpha_d t'' \Delta t_k} - 1}{e^{-\alpha_d \Delta t_k} - 1} \tag{A42}$$

This final equation, $\delta SI_{k,NT}$, can be used with a fitting function to solve for $\alpha_d$, if $\Delta t_k$ is supplied.

## A2.2 Restoration coefficient $\alpha_a$, after a Mistral

To solve for $\alpha_a$, we normalize Eq. (A34) for after a Mistral event, $[t_k + \Delta t_k, t_k + \Delta \tau_k)$, with $\delta SI$ given as:

$$\delta SI_k(t) = \left[ \delta SI_{k-1}^2(t_k) + \frac{D^2}{2} \frac{\delta F_k}{\alpha_d} \left( 1 - e^{\alpha_d \Delta t_k} \right) \right] e^{(\alpha_a - \alpha_d)\Delta t_k - \alpha_a(t - t_k)} \tag{A43}$$

Referencing the time, $t$, to the end of the event, $t''' = t - (t_k + \Delta t_k)$ and plugging $t'''$ into Eq. (A43), we get:

$$\delta SI_k(t''') = \left[ \delta SI_{k-1}^2(t_k) + \frac{D^2}{2} \frac{\delta F_k}{\alpha_d} \left( 1 - e^{\alpha_d \Delta t_k} \right) \right] e^{-\alpha_d \Delta t_k} e^{-\alpha_a t'''} \tag{A44}$$

Normalizing the vertical intercept of $\delta SI_k(t''')$ results in $\delta SI_{k,NI}$:

$$
\begin{aligned}
\delta SI_{k,NI} = \delta SI_k(t''') - \delta SI_k(t''' = 0) &= \left[ \delta SI_{k-1}^2(t_k) + \frac{D^2}{2} \frac{\delta F_k}{\alpha_d} \left( 1 - e^{\alpha_d \Delta t_k} \right) \right] e^{-\alpha_d \Delta t_k} e^{-\alpha_a t'''} \\
&\quad - \left[ \delta SI_{k-1}^2(t_k) + \frac{D^2}{2} \frac{\delta F_k}{\alpha_d} \left( 1 - e^{\alpha_d \Delta t_k} \right) \right] e^{-\alpha_d \Delta t_k} \\
&= \left[ \delta SI_{k-1}^2(t_k) + \frac{D^2}{2} \frac{\delta F_k}{\alpha_d} \left( 1 - e^{\alpha_d \Delta t_k} \right) \right] e^{-\alpha_d \Delta t_k} \left( e^{-\alpha_a t'''} - 1 \right)
\end{aligned}
\tag{A45}
$$

Each post event restratification is normalized to the height of 1 by dividing $\delta SI_{k,NI}$ by $(\delta SI_k(t''' = \Delta \tau_k - \Delta t_k) - \delta SI_k(t''' = $
$0))$:

$$
\begin{aligned}
\delta SI_{k,NH} &= \frac{\delta SI_{k,NI}}{\delta SI_k(t''' = \Delta \tau_k - \Delta t_k) - \delta SI_k(t''' = 0)} \\
&= \frac{\left[ \delta SI_{k-1}^2(t_k) + \frac{D^2}{2} \frac{\delta F_k}{\alpha_d} \left( 1 - e^{\alpha_d \Delta t_k} \right) \right] e^{-\alpha_d \Delta t_k} \left( e^{-\alpha_a t'''} - 1 \right)}{\left[ \delta SI_{k-1}^2(t_k) + \frac{D^2}{2} \frac{\delta F_k}{\alpha_d} \left( 1 - e^{\alpha_d \Delta t_k} \right) \right] e^{-\alpha_d \Delta t_k} \left( e^{-\alpha_a(\Delta \tau_k - \Delta t_k)} - 1 \right)} \\
&= \frac{e^{-\alpha_a t'''} - 1}{e^{-\alpha(\Delta \tau_k - \Delta t_k)} - 1}
\end{aligned}
\tag{A46}
$$

which gives us $\delta SI_{k,NH}$.

To normalize the length of time of post event restratification, we will divide $t'''$ by the post event time length, $\Delta\tau_k - \Delta t_k$, resulting in $t''''$:

$$t'''' = \frac{t'''}{\Delta\tau_k - \Delta t_k} \Rightarrow t''' = t''''(\Delta\tau_k - \Delta t_k) \tag{A47}$$

which leads to $\delta SI_{k,NT}$:

$$\delta SI_{k,NT} = \frac{e^{-\alpha_a t''''(\Delta\tau_k - \Delta t_k)} - 1}{e^{-\alpha_a(\Delta\tau_k - \Delta t_k)} - 1} \tag{A48}$$

This leaves us with an equation of $\alpha_a$, which can be fitted against NEMO model data, if $\Delta t_k$ and $\Delta\tau_k$ are provided.

The average duration and period, $\overline{\Delta t}$ and $\overline{\Delta\tau}$, of events during the preconditioning period in Table 1 are used for $\Delta t$ and $\Delta\tau$ in these normalized equations, Eq. (A42) and (A48). The result of the fitting is shown in Fig. 11, with $\alpha_d$ having a fitted valued of 0.235 $day^{-1}$ and $\alpha_a$ having a fitted value of 0.021 $day^{-1}$.

## A3  Determining $\delta F_k$

With the restoring coefficients determined in the prior section, Sec. A2, and the duration and period of each event available in Table 1, the strength of each Mistral event, $\delta F_k$, can be determined. If we take Eq. (A36) and note the value of $\delta SI_{k-1}$ to be the same as $\delta SI_k(t_k)$ at the beginning of an event, we can simplify the equation in the following steps:

$$\begin{aligned}\delta SI_k(t_k) &= \delta SI_{k-1} \\ \delta SI_k(t_k + \Delta t_k) &= \delta SI_{k-1}e^{-\alpha_d\Delta t_k} + \frac{D^2}{2}\frac{\delta F_k}{\alpha_d}\left(e^{-\alpha_d\Delta t_k} - 1\right)\end{aligned} \tag{A49}$$

And then solve for $\delta F_k$:

$$\delta F_k = \frac{2\left(\delta SI_k(t_k + \Delta t_k) - \delta SI_k(t_k)e^{-\alpha_d\Delta t_k}\right)\alpha_d}{\left(e^{-\alpha_d\Delta t_k} - 1\right)D^2} \tag{A50}$$

The results of $\delta F_k$ for each event in the preconditioning phase is given in Table A1, along with mean and standard deviation.

## A4  Time derivative of $\delta N^2$

Taking the derivative with respect to time of Eq. (A25) and (A34) results in:

$$\frac{\partial\delta N_k^2(t)}{\partial t} = \begin{cases} -\alpha_d\left[\delta N_{k-1}^2(t_k) + \frac{\delta F_k}{\alpha_d}\right]e^{-\alpha_d(t-t_k)} & [t_k, t_k + \Delta t_k) \\ -\alpha_a\left[\delta N_{k-1}^2(t_k) + \frac{\delta F_k}{\alpha_d}\left(1 - e^{\alpha_d\Delta t_k}\right)\right]e^{(\alpha_a - \alpha_d)\Delta t_k - \alpha_a(t-t_k)} & [t_k + \Delta t_k, t_k + \Delta\tau_k) \end{cases} \tag{A51}$$

**Table A1.** The Mistral strengths, $\delta F_k$, for each of the preconditioning phase events, using $\alpha_d$ and $\alpha_a$ from Sec. A2, and the rest of the preconditioning period Mistral characteristics from Table 1, plugged into Eq. (A50).

| Date | $\delta F_k \ s^{-2} day^{-1} \times 10^{-8}$ | Date | $\delta F_k \ s^{-2} day^{-1} \times 10^{-8}$ |
|---|---|---|---|
| 2012-08-30 | 1.81 | 2012-11-27 | 5.46 |
| 2012-09-12 | 2.67 | 2012-12-08 | 6.37 |
| 2012-09-19 | 2.66 | 2012-12-17 | 5.30 |
| 2012-09-28 | 2.73 | 2012-12-21 | 5.03 |
| 2012-10-12 | 3.04 | 2012-12-26 | 4.39 |
| 2012-10-27 | 4.59 | 2013-01-02 | 3.80 |
| 2012-11-11 | 4.84 | 2013-01-23 | 3.53 |
| 2012-11-19 | 3.92 | | |

$\overline{\delta F_k} = 4.01 \times 10^{-8} \ s^{-2} day^{-1}$ and $\sigma_{\delta F_k} = 1.196 \times 10^{-8} \ s^{-2} day^{-1}$

and:

$$\frac{\partial \delta SI_k(t)}{\partial t} = \begin{cases} -\alpha_d \left[ \delta SI_{k-1}(t_k) + \frac{D^2}{2} \frac{\delta F_k}{\alpha_d} \right] e^{-\alpha_d(t-t_k)} & [t_k, t_k + \Delta t_k) \\ -\alpha_a \left[ \delta SI_{k-1}(t_k) + \frac{D^2}{2} \frac{\delta F_k}{\alpha_d} \left( 1 - e^{\alpha_d \Delta t_k} \right) \right] e^{(\alpha_a - \alpha_d)\Delta t_k - \alpha_a(t-t_k)} & [t_k + \Delta t_k, t_k + \Delta \tau_k) \end{cases} \tag{A52}$$

**A5  Asymptotic destratification**

The following sections under Sec. A5 differentiate Eq. (A35) at $t = t_k + \Delta t_k$, or at the end of a Mistral event, where the destratification is the largest, by $k$, and by the other components, $\delta F$, $\Delta t$, and $\Delta \tau$, once $k \to \infty$.

**A5.1  $\frac{\partial \delta SI_k}{\partial k}$**

Equation (A35), at $t = t_k + \Delta t$ results in:

$$\delta SI_k(t_k + \Delta t) = \frac{D^2}{2} \frac{\delta F}{\alpha_d} \left( e^{-\alpha_d \Delta t} - 1 \right) \left( \frac{1 - e^{[(\alpha_a - \alpha_d)\Delta t - \alpha_a \Delta \tau]k}}{1 - e^{(\alpha_a - \alpha_d)\Delta t - \alpha_a \Delta \tau}} \right) \tag{A53}$$

The derivative of Eq. (A53) with respect to (w.r.t.) $k$ is:

$$\frac{\partial \delta SI_k}{\partial k} = \frac{D^2}{2} \frac{\delta F}{\alpha_d} \frac{\left( e^{-\alpha_d \Delta t} - 1 \right)}{\left( 1 - e^{(\alpha_a - \alpha_d)\Delta t - \alpha_a \Delta \tau} \right)} \left( -e^{[(\alpha_a - \alpha_d)\Delta t - \alpha_a \Delta \tau]k} \right) [(\alpha_a - \alpha_d)\Delta t - \alpha_a \Delta \tau] \tag{A54}$$

As $k \to \infty$, with $\alpha_d > \alpha_a$, Eq. (A53) goes to:

$$\delta SI_\infty = \frac{D^2}{2} \frac{\delta F}{\alpha_d} \left( e^{-\alpha_d \Delta t} - 1 \right) \left( \frac{1}{1 - e^{(\alpha_a - \alpha_d)\Delta t - \alpha_a \Delta \tau}} \right) \tag{A55}$$

 **A5.2** $\quad \frac{\partial \delta SI_\infty}{\partial \delta F}$

The derivative of Eq. (A55) w.r.t. $\delta F$ is:

$$\frac{\partial \delta SI_\infty}{\partial \delta F} = \frac{D^2}{2} \frac{1}{\alpha_d} \left( e^{-\alpha_d \Delta t} - 1 \right) \left( \frac{1}{1 - e^{(\alpha_a - \alpha_d)\Delta t - \alpha_a \Delta \tau}} \right) \tag{A56}$$

**A5.3** $\quad \frac{\partial \delta SI_\infty}{\partial \Delta t}$

The derivative of Eq. (A55) w.r.t. $\Delta t$ is:

$$\frac{\partial \delta SI_\infty}{\partial \Delta t} = \frac{D^2}{2} \frac{\delta F}{\alpha_d} \frac{1}{\left( 1 - e^{(\alpha_a - \alpha_d)\Delta t - \alpha_a \Delta \tau} \right)} \left[ -\alpha_d e^{-\alpha_d \Delta t} + \left( e^{-\alpha_d \Delta t} - 1 \right) \left( \frac{(\alpha_a - \alpha_d) e^{(\alpha_a - \alpha_d)\Delta - \alpha_a \Delta \tau}}{1 - e^{(\alpha_a - \alpha_d)\Delta t - \alpha_a \Delta \tau}} \right) \right] \tag{A57}$$

**A5.4** $\quad \frac{\partial \delta SI_\infty}{\partial \Delta \tau}$

The derivative of Eq. (A55) w.r.t. $\Delta \tau$ is:

$$\frac{\partial \delta SI_\infty}{\partial \Delta \tau} = \frac{D^2}{2} \frac{\delta F}{\alpha_d} \left( e^{-\alpha_d \Delta t} - 1 \right) \frac{\left( -\alpha_a e^{(\alpha_a - \alpha_d)\Delta t - \alpha_a \Delta \tau} \right)}{\left( 1 - e^{(\alpha_a - \alpha_d)\Delta t - \alpha_a \Delta \tau} \right)^2} \tag{A58}$$

**Appendix B: 1994 and 2005**

The Mistral event data for the additional two years of 1993-1994 and 2004-2005 are found below in Tab. B1 and B2.

*Author contributions.* Douglas Keller Jr. performed the majority of the analysis presented. Yonatan Givon provided analysis with respect to Mistral events. Romain Pennel provided assistance and guidance running the ocean simulations. Shira Raveh-Rubin and Philippe Drobinski provided analytical assistance and guidance.

*Competing interests.* The authors declare there are no conflicting interests.

*Acknowledgements.* This work is a contribution to the HyMeX program (HYdrological cycle in the Mediterranean EXperiment) through INSU-MISTRALS support and the Med-CORDEX program (COordinated Regional climate Downscaling EXperiment - Mediterranean region). It was also supported by a joint CNRS - Weizmann Institute of Science collaborative project. The authors acknowledge Météo-France for supplying the data and the HyMeX database teams (ESPRI/IPSL and SEDOO/Observatoire Midi-Pyrénées) for their help in accessing the data.

**Table B1.** The start date of, duration of, $\Delta t_k$ $(days)$, and period between each event, $\Delta \tau_k$ $(days)$, for each Mistral event, $k$, and event strength, $\delta F_k$ $(s^{-2} days^{-1})$, for the preconditioning phase of the NEMO simulation period of Jun. 1st, 1993 to May 31st, 1994.

| Date | $\Delta t_k$ | $\Delta \tau_k$ | $\delta F_k$ | Date | $\Delta t_k$ | $\Delta \tau_k$ | $\delta F_k$ |
|---|---|---|---|---|---|---|---|
| 1993-08-28 | 4 | 7 | 9.23e-09 | 1993-12-05 | 1 | 21 | 2.57e-08 |
| 1993-09-04 | 2 | 21 | 8.39e-09 | 1993-12-26 | 3 | 13 | 2.52e-08 |
| 1993-09-25 | 4 | 25 | 1.94e-08 | 1994-01-08 | 1 | 3 | 2.17e-08 |
| 1993-10-20 | 6 | 16 | 2.92e-08 | 1994-01-11 | 2 | 6 | 2.07e-08 |
| 1993-11-05 | 1 | 3 | 2.96e-08 | 1994-01-17 | 15 | 19 | 2.91e-08 |
| 1993-11-08 | 2 | 4 | 2.54e-08 | 1994-02-05 | 7 | 14 | 3.62e-08 |
| 1993-11-12 | 5 | 9 | 2.42e-08 | 1994-02-19 | 1 | 15 | 4.19e-08 |
| 1993-11-21 | 2 | 5 | 2.42e-08 | 1994-03-06 | 3 | 7 | 3.88e-08 |
| 1993-11-26 | 7 | 9 | 2.57e-08 | 1994-03-13 | 6 | 7 | 3.74e-08 |
| 1993-12-05 | 1 | 21 | 2.57e-08 | 1994-03-20 | 3 | 6 | 3.72e-08 |

Preconditioning phase for this year is considered from 1993-08-28 to 1994-04-03 (210 total days).

**Table B2.** Same as Tab. B1, however the NEMO simulation period is from Jun. 1st, 2004 to May 31st, 2005.

| Date | $\Delta t_k$ | $\Delta \tau_k$ | $\delta F_k$ | Date | $\Delta t_k$ | $\Delta \tau_k$ | $\delta F_k$ |
|---|---|---|---|---|---|---|---|
| 2004-09-15 | 2 | 4 | -2.61e-10 | 2004-11-23 | 2 | 4 | 1.93e-08 |
| 2004-09-19 | 10 | 37 | 6.93e-09 | 2004-11-27 | 1 | 7 | 1.88e-08 |
| 2004-10-26 | 1 | 6 | 7.86e-09 | 2004-12-04 | 7 | 14 | 2.30e-08 |
| 2004-11-01 | 2 | 4 | 6.13e-09 | 2004-12-18 | 17 | 31 | 2.30e-08 |
| 2004-11-05 | 16 | 18 | 2.06e-08 | 2005-01-18 | 17 | 24 | 4.10e-08 |
| 2004-11-23 | 2 | 4 | 1.93e-08 | 2005-02-11 | 18 | 21 | 4.43e-08 |

Preconditioning phase for this year is considered from 2004-09-15 to 2005-04-08 (170 total days).

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
