# Peer review of "Untangling the Mistral and seasonal atmospheric forcing driving deep convection in the Gulf of Lion: 2012-2013"

_Ocean Science, 2021_

## Author Comment (AC1)

**Figure 1**

[Figure]

**Figure 2**

[Figure]

**Figure 3**

[Figure]

Filtered wind at 42°E 5°

---

## Author Response (AR1)

We first would like to thank the reviewers for their challenging and insightful comments. They were extremely helpful in discerning where the article needed more clarity and rigor.

**Reviewer 1**

Based on a single deep convection event in the Gulf of Lion, the authors are trying to quantify the Mistral effects to the convection. In particular, they quantified the background contribution, that comes from seasonal changes in the forcing, and the Mistral contribution, that comes as a transient and strong forcing. In my opinion the value of the paper is the use of two approaches for quantification of the above, in particular of development of the "simple" analytical model. On the other side, I feel that the authors invested a lot of energy to basically say that Mistral is generating deep convection events, which is not a new finding and is old as a half of century. Further, saying that some effects comes from seasonality is something to be expected and known, not just in the Gulf of Lion. Summarily, the manuscript can be published, yet after resolving some of protential problems:

1. All the analyses are focused on just one year, which might be characterized with higher or lower deep convection - we do not know how to frame these results into the climatology of the deep convection in the Gulf of Lion. And, there are a great number of studies that are trying to quantify wintertime conditions in the Gulf of Lion at the decadal and climate scales. I don't expect that this might be answered or analysed in this manuscript, however some discussion on that should be placed in discussion and conclusion sections.

We are currently producing a second sister paper attempting to tackle the investigation of multiple years as well. We have added a few sentences to the discussion to clarify that 2012 to 2013 is an above average year in terms of destratification and deep convection.

1. Section 2.2. Why the moving average filter is used, as its transfer function is poor and is allowing some energy on higher frequencies to pass through (up to 20%)? Read e.g. https://ptolemy.berkeley.edu/eecs20/week12/freqResponseRA.html. The leakage of the energy can be even seen in Fig. 2. It would be better to use other filters, like Kaisser Bessel, Butterworth or other.

While it is true that the transfer function for the moving average filter is poor and leaks some energy into the higher frequencies, the way we implemented the moving average intentionally kept the diurnal cycle by averaging along the same point each day. However, it is also true that the moving average isn't the best filter to preserve the low frequency energy either, see the frequency representation of the wind signal here:

[Figure]

The red signal is overall at lower power in the frequency representation, even though it maintains a peak at the diurnal cycle. The blue signal is a Butterworth bandstop filter. At first glance, this filter appears to be a better pick as it more cleanly removes the frequencies between the monthly and daily frequencies (as would be a Kaiser-Bessel or another frequency focused filter). However, our goal is to remove the Mistral signal from the wind (and temperature and humidity, which show similar frequency domain representations) which has some presence in the lower frequencies, which we can see qualitatively with this plot, which covers roughly 3 decades:

[Figure]

And more quantitatively with this plot, that shows the time series of the wind speed with the bandstop, low pass portion of the bandstop, and the moving average filtered signals:

[Figure]

The low frequency energy of the mistral is seen in the low pass signal, particularly around clusters of Mistral events (all Mistral events are shown with the green box overlay). The low frequency muddies the pulse like behavior of the Mistral at times, which is what we saw when we reran the Seasonal simulation with the Butterworth. Both this new Seasonal simulation and the one forced by the moving average filtered forcing had similar results: the pulse like behavior (and the ocean response to said behavior) was still present but less apparent in the Butterworth version (not shown). Because of the low frequency portion of the Mistral's signal found in the cleaner Butterworth filter, we chose the moving average filter because it appeared to remove the Mistral's signal more completely. We admit that this means using the phrase "anomaly timescale" is somewhat of a misnomer, and to account for this we've clarified its use and the presence of the lower frequency energy in the Mistral's signal in the article.

1. Table 1. I don't get why the authors are presenting the length of the period between two Mistral events (tau)? It can be easily estimated from the start date and duration of the event. Also, I am not sure that standard deviation of duration is similar to the standard deviation of in between periods - it does not look from numbers - as presented in the footer of the table.

We just added it for clarity and transparency when looking at the dominating feature of the Mistral, but you are correct it's not necessary. In terms of the similarity of the two standard deviations, we recalculated them and found the same results. Perhaps the larger mean value of the periods obscures this result?

1. Lines 164-168. There should be somewhere the map with locations of CTD measurements.

We added the map, along with the locations of the Argo floats used as well.

1. Section 3.1. There are no comparisons with ARGO data or ocean reanalyses, like MEDSEA which assimilate all the data (including ARGO, which is not used here) through 4D-Var - why?

We added the analysis of Argo floats into the model validation subsection.

**Reviewer 2**

The paper investigates the impact of Mistral events and Mistral event features on deep convection in the Gulf of Lion. A better understanding of the processes of deep convection in the region is very important. The authors show that a meteorological forcing without Mistral pulses is not producing deep convection events in the ocean simulation with NEMO for one seasonal cycle. Additionally they build a toy model which allows studying the impact of Mistral features, which is very interesting. I have some questions and remarks as given in the following.

1. The authors do moving average filtering with a monthly time window of the meteorological forcing data to get rid of the impact of Mistral events in the Seasonal NEMO simulation. I am not sure if they are successful. They give the Mistral event length to 5.6 days and the distance between Mistral events ca. 10 days. In other words, Mistral influences about 1/3 of the month (and more in the investigated winter?). Therefore, I assume that the delta SI in Fig. 7 is only the high-frequency part of the impact of Mistral on the stratification. This might lead to an underestimation of the Mistral impact and favours the Mistral strength over the duration? Why not construct a Mistral-free time series from a longer meteorological time series?

The first part of this question is partially answered by our response to Reviewer 1's 2nd comment, but we will add more here. The moving average filter removes more of the lower frequency portion of the Mistral's signal than a bandstop

Butterworth filter, however it does leave some of the low frequency energy, as can be seen in Dec. 2012 and Jan. 2013 in the wind speed time series plot above. In these cases, yes, we agree that the delta SI primarily represents the high frequency portion of the Mistral. When using the Butterworth filtered forcing to force a new simulation, we see the effect of leaving in more of the lower frequency energy, as commented in our response above, which actually slightly de-emphasizes the Mistral's strength in its pulse like behavior in the delta SI, rather than making it more apparent. This is because the Mistral becomes more frequent and stronger in the winter than the summer (which that shift itself is part of it's lower frequency behavior). Interestingly though, with the moving average filter simulation, removing more of the lower frequency energy emphasizes the pulse like behavior and thus strength of the Mistral.

To answer the second part of the question (why not construct a Mistral free time series), we essentially are doing that with the moving average filter. There a few challenges to making a cleaner time series without the Mistral. Perhaps the biggest challenge is: what do you replace the Mistral event with? We could select the Mistral events from the dates we have (gathered and determined from a larger time series as noted in the article and Givon et al. 2021) and replace it with the moving average but the time series during the preconditioning period would be approximately the same as what we have with the article's moving average filtering of the whole time series. We could instead replace it with the annual average wind speed, but that too would include some of the Mistrals signal through inheritance, as well as potentially introduce low and high frequency artifacts in the signal at the edges of the replaced events, as there would be an enhanced corner in the signal at both the start and end of an event. We considered using a threshold based non-linear filter that would only reduce the wind speed above a certain threshold for Mistral dates, but that too runs into the introduced frequency artifacts from the new corners of the signal at the edges of the events. Given this challenge and the obstacles found within this challenge, we decided the moving average filtering was the most effective given the effort and the results we could obtain from it.

1. Who triggers deep convection? For me, it is more reasonable to blame the seasonality for pre-conditioning and the pulse events, i.e. the Mistral events, as triggering events. Even if there is an accumulation of destratification by Mistral events.

From this study alone, its not clear if the pulse or seasonality triggers deep convection as it appears to be a combination of both. The mixed layer depth grows over the span of a few weeks, which is roughly the same length of time as the length of the Mistral events just preceding the mixed layer reaching the seafloor, but the seasonal change roughly -.2 $m^2/s^2$ in the same span of time (previously Fig. 7). So as both the seasonal SI and delta SI are decreasing during the growth of mixed layer to the seafloor, it's hard to conclusively say for sure. We have adjusted our text to reflect this.

1. Figure 7: it looks like a Mistral event accumulation process and the authors argue that weak Mistral events can even lead to re-stratification. Is it possible, that the system is most sensitive to pulsed disturbances if it is already preconditioned by seasonality? This would fit to the delta SI time series too with an increase in Mistral destratification, followed by a delta SI plateau with smaller changes in the seasonal SI, and a decrease in late spring. And, the simulations are initialised in summer without accumulation but after one seasonal cycle, there is a negative delta SI next summer. So, what about multi-year accumulation?

It is possible that the destratification of the seasonal SI plays a role on the effectiveness of the Mistral's pulse like behavior that is not captured in our simple model, as the model is linear combination of the two with the only connection through the restoration coefficient $\alpha$. And while we can't answer concretely if multi-year accumulation played a role in this article, accumulated stratification would have to overcome by the combined Mistral and seasonal destratification in the next year during the preconditioning phase. We are investigating this in a second sister paper where we are looking at multiple years, trying to determine the causes and effects of the intra-annual variability on deep convection. What we can say though, is that multi-year accumulation has an effect on the salt content and temperature of the dense water formed during deep convection, as was seen for the 2005 deep convection event due to the build up of stratified waters from the lack of deep convection events in the 1990s (Herrmann et al. 2010). We have added a discussion point about it in the article.

1. The authors ask if the Mistral-induced destratification maximum needs to be before the seasonal destratification peak. I assume there is no reason for it and it is just like that in the chosen period. In another season such an extreme Mistral outlier might easily be later in the plateau period. So, why not more years investigated? It seems not to be very difficult.

This is our line of thinking as well. Our next investigation is looking into how the timing of the Mistral preconditioning and the seasonal preconditioning changes over multiple years. We decided it was best to organize the development of the simple model and multi-year investigation into multiple papers rather than try to fit it all into one.

1. line 34: here, in the introduction, the main conclusion is given already. I suggest skipping it here.

We removed the concluding comment from the introduction.

1. line 74: the heat (and kinetic energy) flux by rainfall is considered?

The heat flux, but not the kinetic energy flux, is calculated in the bulk formulation and considered in the NEMO simulation but not during the buoyancy loss analysis. The effects from the water flux are almost negligible (Somot 2016). We've added text in the article when presenting the bulk formulas to clarify this.

1. line 89: the year in one reference is missing.

Fixed.

1. line 110: intra-day variability is misleading. Only, the average diurnal cycle is kept as is clarified later in the text.

Changed to "average intra-day variability" to be more clear and accurate.

1. line 188: Can it be something else besides Mistral events?

Possibly but unlikely. On the atmospheric side, the Mistral is the primary phenomenon acting on that timescale causing buoyancy loss in the Gulf of Lion during the preconditioning period. On the ocean side, most phenomena act on too long of timescales to influence the roughly weekly timescale, so to our knowledge it's the Mistral.

1. line 390: next -> net

Fixed.

1. Figure 6: the box is difficult to be seen

Increased the box line width to make it more visible.